# Inhibition of the *de novo* pyrimidine biosynthesis pathway limits ribosomal RNA transcription causing nucleolar stress in glioblastoma cells

M. Carmen Lafita-Navarro[1], Niranjan Venkateswaran[1], Jessica A. Kilgore[2], Suman Kanji[3], Jungsoo Han[4], Spencer Barnes[5], Noelle S. Williams[2], Michael Buszczak[4], Sandeep Burma[3,6], Maralice Conacci-Sorrell[1,7,8]*

1 Department of Cell Biology, University of Texas Southwestern Medical Center, Dallas, Texas, United States of America, 2 Department of Biochemistry, University of Texas Southwestern Medical Center, Dallas, Texas, United States of America, 3 Department of Neurosurgery, University of Texas Health Science Center, San Antonio, Texas, United States of America, 4 Department of Molecular Biology, University of Texas Southwestern Medical Center, Dallas, Texas, United States of America, 5 Bioinformatics Core Facility, University of Texas Southwestern Medical Center, Dallas, Texas, United States of America, 6 Department of Biochemistry and Structural Biology, University of Texas Health Science Center, San Antonio, Texas, United States of America, 7 Hamon Center for Regenerative Science and Medicine, University of Texas Southwestern Medical Center, Dallas, TX, United States of America, 8 Harold C. Simmons Comprehensive Cancer Center, University of Texas Southwestern Medical Center, Dallas, Texas, United States of America

* Maralice.ConacciSorrell@UTSouthwestern.edu

**Data Availability Statement:** All relevant data are within the manuscript and its Supporting Information files.

## Abstract

Glioblastoma is the most common and aggressive type of cancer in the brain; its poor prognosis is often marked by reoccurrence due to resistance to the chemotherapeutic agent temozolomide, which is triggered by an increase in the expression of DNA repair enzymes such as MGMT. The poor prognosis and limited therapeutic options led to studies targeted at understanding specific vulnerabilities of glioblastoma cells. Metabolic adaptations leading to increased synthesis of nucleotides by *de novo* biosynthesis pathways are emerging as key alterations driving glioblastoma growth. In this study, we show that enzymes necessary for the *de novo* biosynthesis of pyrimidines, DHODH and UMPS, are elevated in high grade gliomas and in glioblastoma cell lines. We demonstrate that DHODH's activity is necessary to maintain ribosomal DNA transcription (rDNA). Pharmacological inhibition of DHODH with the specific inhibitors brequinar or ML390 effectively depleted the pool of pyrimidines in glioblastoma cells grown *in vitro* and *in vivo* and impaired rDNA transcription, leading to nucleolar stress. Nucleolar stress was visualized by the aberrant redistribution of the transcription factor UBF and the nucleolar organizer nucleophosmin 1 (NPM1), as well as the stabilization of the transcription factor p53. Moreover, DHODH inhibition decreased the proliferation of glioblastoma cells, including temozolomide-resistant cells. Importantly, the addition of exogenous uridine, which reconstitutes the cellular pool of pyrimidine by the salvage pathway, to the culture media recovered the impaired rDNA transcription, nucleolar morphology, p53 levels, and proliferation of glioblastoma cells caused by the DHODH inhibitors. Our *in vivo* data indicate that while inhibition of DHODH caused a dramatic reduction in pyrimidines in tumor cells, it did not affect the overall pyrimidine levels in normal brain and liver tissues,

**Funding:** This research was supported by Cancer Prevention and Research Institute of Texas RR150059 and RP150596, American Cancer Society IRG-17-174-13, Welch I-1914, NCI R01CA245548, University of Texas Southwestern Circle of Friends Early Investigator Award to M.C-S, the NIH (R01GM125812) to M.B, and NIH grant CA197796, NNX16AD78G to SB, 1S10OD021684-01 award to Dr. Kate Luby-Phelps. SB is the Mays Family Foundation Distinguished Chair in Oncology. M.C-S is the Virginia Murchison Linthicum Scholar in Medical Research. The funders had no role in study design, data collection and analysis, decision to publish, or preparation of the manuscript.

**Competing interests:** The authors have declared that no competing interests exist.

suggesting that pyrimidine production by the salvage pathway may play an important role in maintaining these nucleotides in normal cells. Our study demonstrates that glioblastoma cells heavily rely on the *de novo* pyrimidine biosynthesis pathway to generate ribosomal RNA (rRNA) and thus, we identified an approach to inhibit ribosome production and consequently the proliferation of glioblastoma cells through the specific inhibition of the *de novo* pyrimidine biosynthesis pathway.

## Author summary

The current standard therapy for glioblastoma, the most malignant brain tumor, was established more than a decade ago and relies on a combination of surgery, radiation, and the DNA methylating agent temozolomide. Here, we report a new approach to target glioblastoma growth through the inhibition of the *de novo* biosynthesis of pyrimidines, which preferentially limits ribosomal RNA (rRNA) production. Cancer cells have elevated rates of rRNA synthesis so that they can produce enough ribosomes to meet the demands for protein synthesis that are linked to increase cell growth and division. Therefore, targeting aberrant rRNA production by reducing nucleotide availability could provide an effective strategy to treat glioblastoma and, potentially, other tumor types.

## Introduction

Tumors activate purine and pyrimidine biosynthetic pathways to increase the supply of nucleotides to fulfill the requirements of highly proliferative cells [1, 2]. Purines, nucleotides with adenine and guanine bases, and pyrimidines, with uracil, cytosine, or thymine bases, are necessary for the synthesis of RNA, DNA, nucleotide-activated sugars, and lipids [3]. Both purines and pyrimidines can be synthesized by 2 alterative pathways: the *de novo* pathways that metabolize ribose and amino acids in a series of enzymatic reactions and the salvage pathways that recycle nucleotides present in the cells or their environment through phosphorylation/dephosphorylation reactions [3].

The 6 enzymatic reactions of the *de novo* pyrimidine biosynthesis pathway are performed by 3 essential enzymes: 1-Carbamoyl-Phosphate Synthetase 2, 2-Aspartate Transcarbamylase, and 3-Dihydroorotase (CAD); Dihydroorotate dehydrogenase (quinone) (DHODH); and 1-Orotate Phosphoribosyl Transferase and 2-Orotidine-5'-Decarboxylase/Uridine Monophosphate Synthetase (UMPS). CAD carries out the 3 first enzymatic reactions where glutamine and bicarbonate are sequentially processed into carbamoyl-phosphate, carbonyl-L-aspartate and dihydroorotate [3]. DHODH, the only enzyme in this pathway that localizes in the inner mitochondrial membrane, mediates the oxidation of dihydroorotate to orotate by reducing ubiquinone [4]. In the mitochondria, DHODH also interacts with respiratory complexes II and III [5]; thus, DHODH's activity is necessary for mitochondrial electron transport chain function in addition to the production of pyrimidines [3]. UMPS carries out the 2 last steps of the *de novo* pyrimidine biosynthetic pathway by producing orotidine monophosphate (OMP) from orotate and the pentose phosphate phosphoribosyl pyrophosphate (PRPP) through its orotate phosphoribosyl transferase activity, and uridine monophosphate (UMP) by decarboxylation of OMP [3]. UMP, which is used to synthesize UDP, UTP, dTTP, CTP, and dCTP, is the common metabolite between the *de novo* and the salvage pyrimidine synthesis pathway. In the salvage pathway, UMP can be generated by the conversion of uridine (already present in

cells or taken up from the environment) into UMP by the activity of uridine/cytidine kinase 1/2 (UCK1/2) [6].

The genes encoding the *de novo* pyrimidine biosynthesis enzymes are upregulated in highly proliferative cells. The oncogene MYC upregulates the expression of *CAD* and *DHODH* [7, 8]. Moreover, our lab recently discovered that the transcription factor aryl hydrocarbon receptor (AHR), which is necessary for glioblastoma growth [9–11], mediates the expression of *DHODH* and *UMPS* in MYC-overexpressing fibroblasts and glioblastoma cells [12]. Another mechanism used by cancer cells to stimulate the *de novo* pyrimidine synthesis pathway is the activation of CAD by phosphorylation [13]. Importantly, different cancer types, such as acute and chronic myeloid leukemia (AML and CML) [14, 15], multiple myeloma [16], melanoma [17], K-Ras–driven pancreatic cancer [18], PTEN-mutated breast cancer or glioblastoma [19], and chemotherapy-resistant triple-negative breast cancer cells [13], have been found to be vulnerable to the inhibition of DHODH.

Cancer cells augment the production of ribosomes to ensure continuous protein synthesis for efficient cell growth. rRNAs, which constitute about 80% of the total cellular RNA [20], are essential components of ribosomes and thus necessary for protein synthesis. There are 4 mature rRNAs, 18S, 5.8S and 28S, which are derived from the precursor 47S pre-rRNAs transcribed of rDNA loci by RNA polymerase I; and the 5S rRNA, synthesized by RNA polymerase III [21]. There are about 400 rDNA loci per diploid genome in human cells and about 20–50% of them are transcriptionally active [22], ensuring sufficient rRNA is produced for ribosome biogenesis [21]. The different steps of ribosome production (rDNA transcription, pre-rRNA processing, and assembly of the ribosome subunits) occur in the nucleolus [23]. Due to its involvement in ribosome biogenesis, the morphology, size, and/or number of nucleoli vary according to their functional state. In fact, the size of the nucleolus positively correlates with rRNA synthesis, cell growth, and malignancy of tumors [24].

Glioblastomas are grade IV gliomas and the most frequent malignant tumors of the brain. According to the Central Brain Tumor Registry of the United States, based on data collected from 2001 to 2015, glioblastomas were mainly diagnosed in patients with an average age of 65 years old [25]. However, children and younger adults can also develop glioblastoma tumors. The survival rate for patients diagnosed with glioblastoma varies by age with an overall 5-year survival rate around 6.8% [25]. Despite this low survival rate, the therapy for glioblastoma has not improved significantly since 2005 when the DNA methylating agent temozolomide was incorporated to the standard therapy regimen of surgical resection followed by radiotherapy [26, 27]. Glioblastomas grow very rapidly and frequently develop resistance to treatment by increasing the expression of DNA repair enzymes such as O-6-Methylguanine-DNA Methyltransferase (MGMT), thus explaining in part the very poor prognosis [28–30]. The lack of therapeutic options for treatment of glioblastoma patients have created an interest in identifying metabolic dependencies of glioblastoma cells that can be specifically manipulated [31–33].

Alterations that activate the *de novo* nucleotide biosynthesis pathways are emerging as key features of glioblastomas [34–37] and have been shown to be essential to maintain the stemness of glioblastoma-initiating cells and tumor growth [34–36]. Recently, inhibition of the purine guanosine monophosphate biosynthesis was shown to decrease the production of rRNA and glioblastoma cell growth [36], and to increase the sensitivity to radiotherapy [37]. Here, we demonstrate that glioblastoma cells rely on the *de novo* biosynthesis of pyrimidines to support rDNA transcription and thus cell growth. Inhibition of DHODH, which is necessary for the *de novo* synthesis of pyrimidines, does not impair cell growth or pyrimidines abundance in normal cells and tissues, indicating that they do not depend entirely on the *de novo* pyrimidine synthesis pathway for the synthesis of pyrimidines, rRNA production, and proliferation. Thus, we report here that glioblastoma cells, including TMZ-resistant cells, are

specifically vulnerable to pharmacologic inhibition of the enzyme DHODH and that inhibiting *de novo* pyrimidine biosynthesis effectively decreases the production of rRNA in glioblastoma cells.

## Results

### Inhibition of *de novo* pyrimidine biosynthesis decreases rRNA transcription in glioblastoma cells

By analyzing The Cancer Genome Atlas (TCGA) (Fig 1A) and the Chinese Glioma Genome Atlas (CGGA) (S1A Fig), we found that the mRNA levels of the enzymes in the *de novo* pyrimidine pathway, *DHODH* and *UMPS*, were elevated in high-grade glioma (IV/glioblastoma) patient samples. To investigate the need of glioblastoma cells for the *de novo* pyrimidine biosynthesis pathway, we chose 3 distinct cell lines (S1B Fig): SF188, a commercially available pediatric male cell line expressing p53 that is mutated in the DNA binding domain (G622E); LN229, a commercially available adult female cell line with mutated p53 outside the DNA binding domain (P98L); and the patient-derived GBM9 with unknown p53 status [38].

Western blot analysis indicated that DHODH and UMPS protein levels were higher in the glioblastoma cells LN229, GBM9, and SF188 in comparison to normal human p14ARF-/- immortalized astrocytes, which are non-transformed differentiated glial cells (Fig 1B) [27]. The high expression of these enzymes in glioblastoma cells, suggests a greater reliance on the *de novo* pyrimidine biosynthetic pathway for the development and/or progression of glioblastoma (IV grade glioma), which agrees with data from recently published studies [34–36].

The pyrimidines UTP and CTP are necessary for RNA synthesis; therefore, we reasoned that transcriptionally hyperactive sites such as rDNA loci would be more dramatically affected by a decrease in the supply of intracellular pyrimidines. Given that glioblastoma cells exhibited increased levels of the enzymes involved in *de novo* pyrimidine biosynthesis, we hypothesized that rRNA production in glioblastoma cells would be altered by inhibitors of the *de novo* pyrimidine biosynthesis enzymes (Fig 1C and 1D). To test this hypothesis, we used 2 DHODH inhibitors, brequinar, which is currently in clinical trials for leukemia (ClinicalTrials.gov Identifier: NCT03760666) and ML390 (Fig 1C) [39]. To confirm that brequinar and ML390 specifically inhibit the *de novo* pyrimidine biosynthesis in glioblastoma cells, we measured the concentration of UMP, UDP, UTP, and uridine by LC-MS/MS in LN229 and GBM9 cells treated with brequinar and ML390. These analyses showed that brequinar and ML390 effectively lowered the concentration of UTP, UDP, UMP, and uridine in treated cells (Fig 1E and 1F, and S1C Fig).

To determine the effects of limiting the pyrimidine pool on rRNA transcription, we added brequinar or ML390 to the culture media of LN229, GBM9 and SF188 cells and studied their effects on the production of 47S pre-rRNA as measured by qPCR. These assays were performed in the presence or absence of uridine, which should rescue the inhibitory effects of brequinar and ML390 due to its ability to reconstitute the pyrimidine pool via the salvage pathway (Fig 1C). As predicted, inhibition of DHODH with brequinar or ML390 led to a decrease in 47S pre-rRNA in all cell lines tested (Fig 1G and 1H), and the addition of uridine to the media rescued these effects. These results indicate that the decrease of pre-rRNA upon brequinar or ML390 treatment was indeed due to a reduction in pyrimidines supply (Fig 1G and 1H). Interestingly, *ACTIN* mRNA was not affected upon inhibition of the *de novo* pyrimidine biosynthesis pathway (Fig 1I and 1J), suggesting that highly active transcriptional sites such as rDNA may be more sensitive to decreases in intracellular pyrimidines.

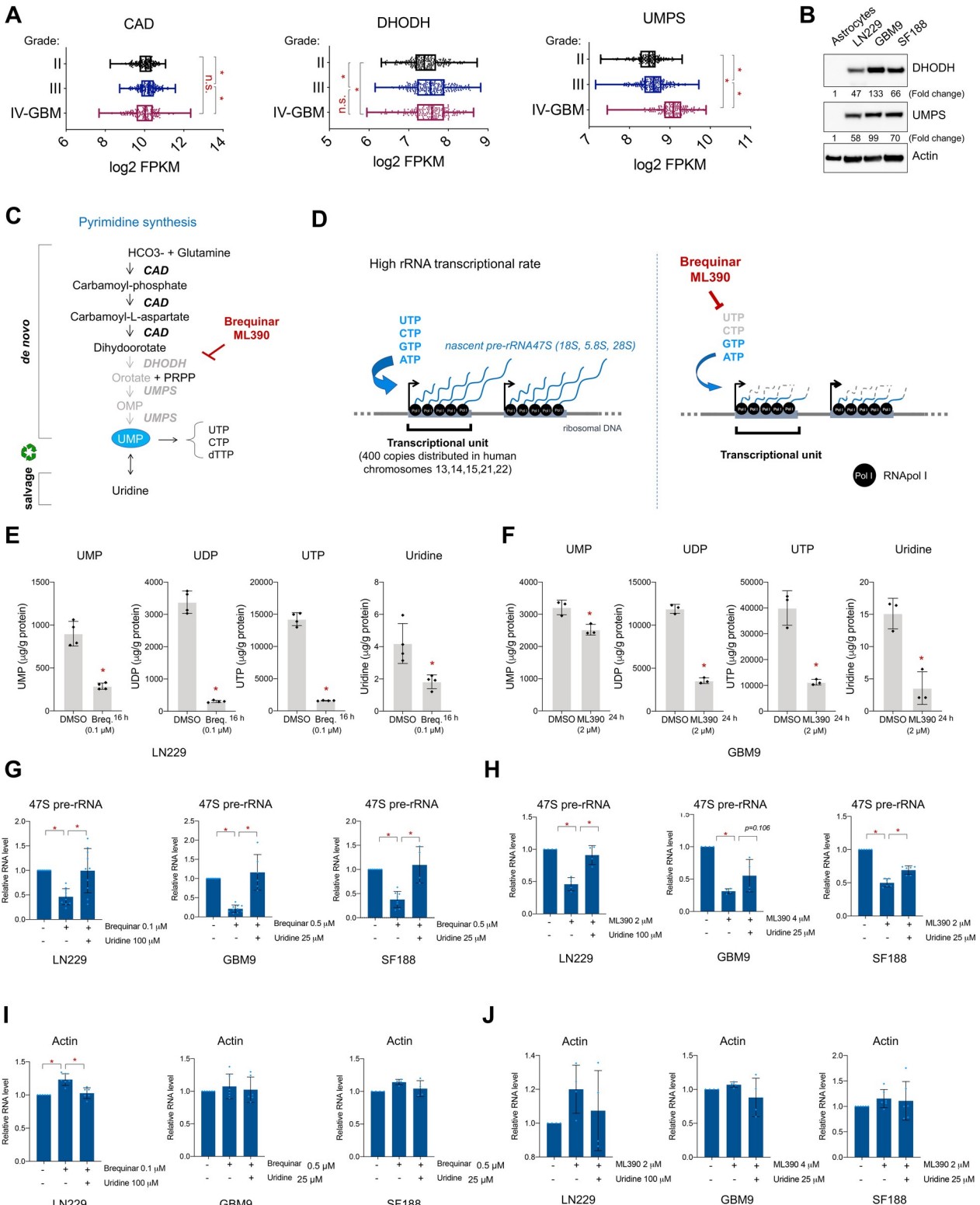

**Fig 1. *De novo* pyrimidine biosynthesis is necessary to maintain efficient ribosomal RNA transcription in glioblastoma.** (A) *CAD*, *DHODH* and *UMPS* mRNA levels in lower (II-III) to higher grade (IV/GBM) glioma patients from TCGA database. (B) Western blot of p14 ARF-/- human astrocytes, LN229, GBM9, and SF188. Fold change in expression as indicated was normalized to actin levels and compared to the expression in astrocytes by densitometry analysis with Image J. (C) Representation of *de novo* and salvage pyrimidine biosynthesis pathways and brequinar. (D) Representation of the high rDNA transcription rate (left panel) and the potential effects of blocking the *de novo* pyrimidines biosynthesis (right

panel). (E-F) UMP, UDP, UTP and uridine measured by LC-MS/MS in LN229 after incubation with brequinar, 4 replicates, (E) and in GMB9 after incubation with ML390 (F), 3 replicates. (G) qPCR of 47S pre-rRNA with or without brequinar and uridine for 24 h in LN229, GBM9, and SF188. N = 2–5. (H) qPCR of 47S pre-rRNA with or without ML390 and uridine for 48 h in LN229, GBM9, and SF188. N = 2–3. (I-J) qPCR of *ACTIN* with or without brequinar for 24 h (I) or ML390 (J) for 48 h and uridine in LN229, GBM9, and SF188. N = 2–4. For G-J, individual technical replicates values of each biological replicate are represented by diamonds. * indicates p-values ≦0.05. Numerical values for each of the experiments represented are available in S1 Data.

## Inhibition of the *de novo* pyrimidine biosynthesis pathway decreases proliferation of glioblastoma cells *in vitro*

Given the effects of brequinar and ML390 on the production of 47S pre-rRNA, we asked whether prolonged inhibition of DHODH affects the proliferation of glioblastoma cells. First, we measured the proliferation of non-transformed cells (human p14ARF-/- astrocytes (Fig 2A) and retinal epithelial ARPE cells (S2A Fig)), and the glioblastoma cells LN229, GBM9, and SF188 over 6 days in the presence of increasing amounts of brequinar (Fig 2A) or ML390 (S2A and S2B Fig). Concentration ranges of brequinar and ML390 used to treat glioblastoma cells were selected based on previous studies in leukemia cells [14]. Glioblastoma cells were more sensitive to brequinar and ML390 than the human astrocytes (Fig 2A) or ARPE (S2A Fig), indicating that these inhibitors specifically affect proliferation of transformed glioblastoma cells.

To test whether addition of exogenous uridine can rescue the effects of inhibiting DHODH in proliferation, we first determined the highest uridine concentrations tolerated by glioblastoma cells without causing off-target effects and cell death (S2C Fig). We then chose non-toxic concentrations of uridine (100 μM for LN229, and 10 and 25 μM for GMB9 and SF188) for the rescue experiments. Brequinar decreased proliferation of LN229, GBM9, and SF188 cells but not ARPE. The addition of uridine to the culture media rescued growth of glioblastoma cells (Fig 2B and S2D Fig). Importantly, uridine did not increase proliferation of the DMSO-treated cells (Fig 2B and S2C and S2D Fig). Altogether, these results indicate that uridine specifically rescues the inhibition of DHODH and the growth of glioblastoma cells.

Consistent with the effects of the DHODH inhibitors, knocking down *DHODH* expression by siRNA in LN229 and GBM9 cells decreased their proliferation. However, the addition of uridine to the media did not rescue this effect (S3A and S3B Fig). Similar results were previously reported in HeLa cells [5]. Moreover, knocking down *DHODH* by siRNA, did not abolish DHODH expression (S3A and S3D Fig), and it was not sufficient to decrease the levels of pyrimidines (S3E Fig) or the abundance of 47S pre-rRNA (S3C Fig). Importantly, treating cells with low concentrations of brequinar (0.01 μM) (S3E Fig) was not sufficient to reduce the levels of UMP, UDP, UTP and uridine compared with 0.1 μM that was used in all other experiments (Fig 1E). It is possible that the residual DHODH levels after siRNA transfection are enough to maintain the production of pyrimidines through the *de novo* pathway. Moreover, because DHODH is a structural mitochondrial protein also involved in the electron transport chain, the effects of *DHODH* knockdown on proliferation (S3A and S3B Fig.) could be also due to alterations in mitochondrial activity as previously shown [4, 5].

To eliminate the possibility that the decrease in any nucleotide type leads to a decrease in 47S pre-rRNA synthesis due to cell cycle arrest, we used 5-fluorouracil (5-FU), an inhibitor of the enzyme thymidylate synthetase (TS), which produces dTMP from dUMP (S4A Fig). As expected, 5-FU decreased proliferation of glioblastoma cells (S4B and S4E Fig), and the addition of uridine did not rescue this effect (S4E Fig). Furthermore, as dTTP is not incorporated into rRNA, 5-FU did not affect the production of the 47S pre-rRNA (S4C and S4D Fig) as previously shown [40]. Therefore, we conclude that DHODH inhibition specifically limits the production of pre-rRNA and thus affects the viability of glioblastoma cells.

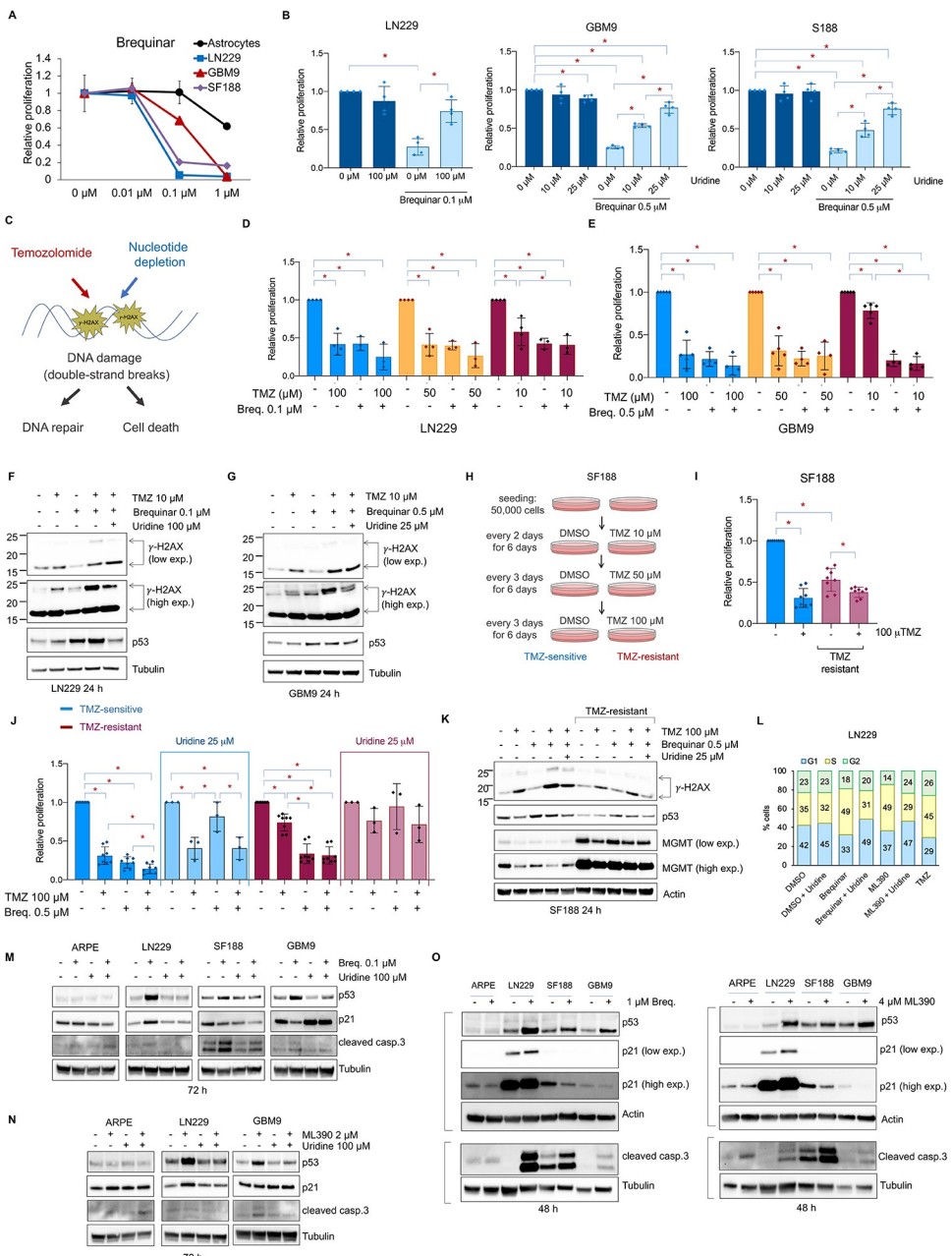

**Fig 2. Inhibition of the *de novo* pyrimidine biosynthesis pathway decreases proliferation of temozolomide-sensitive and -resistant glioblasomta cells.** (A) Relative proliferation of p14 ARF-/- human astrocytes, LN229, GBM9 and SF188 cells with increasing amounts of brequinar. Media with drugs was replaced every 2 days for 6 days. N = 3. (B) Relative proliferation of LN229, GBM9 and SF188 cells with or without brequinar and uridine. Media with drugs was replaced every 2 days for 6 days. N = 3–4. (C) Representation of the effects of temozolomide (TMZ) and the depletion of nucleotides causing DNA double-strand breaks and the phosphorylation of H2AX ($\gamma$-H2AX). (D, E) Relative proliferation of LN229 and GBM9 in the presence of TMZ, brequinar (Breq.) or brequinar + TMZ. Media with drugs was replaced the day after seeding, and cell proliferation was measured 4 days later. N = 3–4. (F, G) Western blot of LN229 and GBM9 for $\gamma$-H2AX and p53. Also see Supplementary S2F and S2G Fig. (H) Representation of the *in vitro* generation of SF188 TMZ-resistant cells. (I) Relative proliferation of SF188 TMZ-sensitive and -resistant cells with or without TMZ. (J) Relative proliferation of SF188 TMZ-sensitive or -resistant cells with or without TMZ, brequinar or brequinar + TMZ with or without uridine normalize to each DMSO condition. Also see Supplementary S2H Fig. Media with drugs was replaced the day after seeding, and cell proliferation was measured 4 days later. (K) Western blot of SF188 TMZ-sensitive or -resistant cells with or without TMZ, brequinar or brequinar + TMZ with or without uridine for $\gamma$-H2AX, p53 and, MGMT. Also see Supplementary S2I and S2J Fig. (L) Cell cycle analysis of

LN229 treated with 0.1 µM brequinar, 2 µM ML390 in the presence or absence of 100 µM uridine, and 100 µM TMZ for 24 h. Media with drugs was replaced the day after seeding and cells harvested after 24 h. Also see Supplementary S2K Fig. This experiment was done three times with similar results. (M) Western blot of ARPE, LN229, SF188 and GBM9 for p53, p21 and cleaved caspase 3 after 72 h of treatment with 0.1 µM brequinar in the presence or absence of 100 µM uridine. (N) Western blot of ARPE, LN229, and GBM9 for p53, p21 and cleaved caspase 3 after 72 h of treatment with 2 µM ML390 in the presence or absence of uridine 100 µM. (O) Western blot of ARPE, LN229, SF188 and GBM9 for p53, p21 and cleaved caspase 3 after 48 h of treatment with 1 µM brequinar or 4 µM ML390. For all the Western blot experiments, media with drugs and metabolites were replaced the day after seeding and cells harvested at the indicated time points. Anti-phosphorylated H2AX antibody shows non-ubiquitinated (~ 15 KDa) and ubiquitinated (~ 25 KDa) $\gamma$-H2AX. * indicates p-values ≦0.05. Numerical values for each of the experiments represented are available in S2 Data.

## Inhibition of the *de novo* pyrimidine synthesis decreases proliferation of temozolomide- resistant glioblastoma cells

TMZ is the main chemotherapeutic agent used for the treatment of glioblastoma [29]. TMZ is an alkylating agent that methylates adenine and guanine residues in DNA, leading to DNA damage and double-strand breaks (Fig 2C) [29]. DNA double-strand breaks are marked by S139 phosphorylation of H2AX (known as $\gamma$-H2AX) [41]. Interestingly, pyrimidine depletion has been shown to also cause cellular stress, DNA damage, and double-strand breaks (Fig 2C) [42]. Therefore, we asked whether the combination of TMZ with brequinar synergized to kill glioblastoma cells. LN229 and GBM9 cells were highly sensitive to TMZ (Fig 2D and 2E, and S2E Fig), and we did not observe synergistic effects on proliferation when TMZ was combined with brequinar (Fig 2D and 2E). Treating LN229 and GBM9 cells with 10 µM TMZ, which had a limited effect on proliferation (S2E Fig), with brequinar or with a combination of both agents in the presence or absence of uridine, demonstrated that TMZ, but not brequinar, increased H2AX phosphorylation (Fig 2F and 2G and S2F and S2G Fig). However, brequinar potentiated the effect of TMZ on the phosphorylation of H2AX, and this increase was reduced by the addition of uridine (Fig 2F and 2G and S2F and S2G Fig). These results suggest that the inhibition of the *de novo* pyrimidine biosynthesis pathway enhance the DNA damage induced by TMZ in glioblastoma cells.

Previously, it was shown that SF188 cells were less sensitive to TMZ than other glioblastoma cells [43]. Yet SF188 cell proliferation was still reduced when treated with 100 µM TMZ (Fig 2I). We, therefore, generated a TMZ-resistant SF188 cell line by treating cells for approximately 3 weeks with increasing concentrations of TMZ until the population growth was resistant to the addition of 100 µM TMZ (Fig 2H and 2I). TMZ-resistant cells grew slower than the TMZ-sensitive cells even in the absence of TMZ (Fig 2I, and S2H Fig). We then compared the effects of TMZ and brequinar alone or in combination on the proliferation of the TMZ-sensitive and TMZ-resistant cells in the presence or absence of uridine. The TMZ-sensitive cells showed decreased proliferation in the presence of TMZ or brequinar alone and the combination of both agents decreased it further (Figs 2J and S2H). The effects of brequinar alone or in combination with TMZ were rescued by the addition of uridine to the media (Fig 2J, and S2H Fig). Importantly, TMZ-resistant cells were still sensitive to brequinar and responsive to the addition of uridine (Fig 2J and S2H Fig). Furthermore, we compared the effects of TMZ and brequinar on DNA damage in the TMZ-sensitive and TMZ-resistant SF188 cells. We found that TMZ, but not brequinar, increased H2AX phosphorylation. This increase was enhanced when TMZ and brequinar were combined, and this effect was rescued by the addition of uridine (Fig 2K, and S2I Fig).

The TMZ-resistant cells showed decreased basal levels of $\gamma$-H2AX, which is in agreement with the acquisition of TMZ resistance in glioblastoma that is due to an increase in the levels of DNA repair enzymes such as MGMT (Fig 2K) [28]. Interestingly, our results show that

TMZ as well as brequinar alone decreased the levels of MGMT in the TMZ-sensitive SF188 cells. This effect was more dramatic when both agents were combined (Fig 2K and S2J Fig). It has been shown that TMZ, in addition to causing direct DNA damage, limits repair by down-regulating MGMT expression in some glioblastoma cell lines [44, 45]. This together with the reduction in MGMT expression caused by brequinar explains the increased phosphorylation of H2AX in the combined TMZ and brequinar treatment (Fig 2F, 2G and 2K, and S2J Fig).

To determine whether the effects of TMZ, brequinar or ML390 on proliferation were cyto-static or cytotoxic, we treated LN229 cells with DMSO, 0.1 μM brequinar, 2 μM ML390 (in the absence or presence of 100 μM uridine) and with 100 μM TMZ for 24 h and analyzed the cell cycle by flow cytometry. Brequinar and ML390 arrested the cell cycle in S phase, and this effect was rescued in the presence of uridine (Fig 2L and S2K Fig), which is in agreement with previous results [46, 47]. This result suggests that the depletion of pyrimidine nucleotides not only impairs the ability of cells to produce rRNA, but also, as expected, it impairs the ability of the cells to synthesize DNA and progress through the cell cycle. TMZ caused cell cycle arrest in S/G2 phase (Fig 2L and S2K Fig), as shown by others [48, 49], which is in line with the DNA damage checkpoint response at the end of the G2 phase [50]. Interestingly, in the conditions described above, there was no measurable cytotoxicity for any of the 3 drugs as indicated by the absence of subG0 cell cycle phase.

## p53 protein levels are elevated upon treatment with DHODH inhibitors in glioblastoma cells

The transcription factor and tumor suppressor p53 senses DNA damage, leading to its protein stabilization and the induction of its target genes [51]. Because TMZ and its combination with brequinar increased DNA damage, we measured p53 protein levels in cells treated with brequinar, ML390, and/or TMZ. The concentrations of TMZ (10 μM for LN229 and GMB9, and 100 μM for SF188) used in this study did not cause an increase p53 levels in the tested glioblastoma cell lines. This is contrary to what was previously shown in glioblastoma, neuroblastoma and melanoma cells [52–55]. Surprisingly, treatment with ML390 or brequinar, which did not increase DNA damage (Fig 2F, 2G and 2K), did cause an increase in p53 protein levels that was reversed by the addition of uridine (Fig 2F, 2G, 2K and 2M–2O, and S2F, S2G and S2J Fig). These results indicate that the inhibition of the *de novo* pyrimidine biosynthesis pathway, but not DNA damage, are responsible for the increase in p53 observed in cells treated with brequinar or ML390 (Fig 2G, 2H, 2K and 2M–2O).

One of the best-known transcriptional targets of p53 is the cell cycle inhibitor p21 [56]. We examined p21 expression as a readout of p53 activity and found that despite the increase p53 levels upon brequinar or ML390 treatment in all the glioblastoma cell lines, p21 only increased in LN229 (Fig 2M–2O). This result suggests that p53 is transcriptionally inactive in SF188 and GBM9 cells. This is consistent with the p53 status in SF188 (S1B Fig.), which is mutated in the DNA binding domain (G266E). The status of p53 in GBM9 cells is not known. However, given that p21 did not increase even when p53 protein levels did, our results suggest that p53 is not active in GBM9. LN229 cells express a mutated p53 (P98L) form (S1B Fig) that does not affect the DNA binding domain of p53 and remains capable of activating p21 in this cell line.

p21 expression was not elevated in SF188 and GBM9 cells upon DHODH inhibition, indicating that additional mechanisms are responsible for reduced proliferation in these cells upon brequinar and ML390 treatment. Therefore, we examined apoptosis by measuring cleaved caspase 3 to define whether the decrease in cell number upon DHODH inhibition in glioblastoma cells was also due to increase cell death. Treatment with 0.1 μM brequinar activated apoptosis in SF188 and to a lesser extent in GBM9, but not in ARPE or LN229 (Fig 2M). Similarly,

treatment with 2 μM ML390 activated apoptosis in GBM9 but not in ARPE or LN229 (Fig 2N). However, higher concentrations of brequinar and ML390 (1 μM and 4 μM, respectively), activated apoptosis in LN229, SF188 and GBM9, but not in ARPE. Altogether, these results suggest that brequinar and ML390 cause a combination of cytostatic and cytotoxic effects, leading to impaired cell proliferation specifically in glioblastoma cells.

## Inhibition of the *de novo* pyrimidine biosynthesis decreases growth and rRNA production in LN229 tumor xenografts

To determine whether the inhibition of the *de novo* pyrimidine biosynthesis regulates cell growth and rRNA production *in vivo*, we performed subcutaneous xenografts in NOD-S-CID mice using the glioblastoma cell line LN229. LN229 cells ($1 \times 10^6$) were injected into one flank of each mouse. After 6 days, 15 mg/kg brequinar sodium was injected intraperitoneally (IP) every 3 days for a period of 60 days, according to a previously published protocol for brequinar treatment *in vivo* [14]. Although this regimen was effective in reducing malignancy in leukemia [14], in the LN229 tumor xenografts, the reduction of tumor size was not significant (S5A Fig). Therefore, we performed another subcutaneous xenograft experiment, injecting $3 \times 10^6$ LN229 cells and starting treatment 5 days after injection with a daily regimen of 10 mg/kg sodium brequinar for 55 days (Fig 3A). Once the tumor volume reached 100 mm$^3$, tumor volumes were periodically recorded. Overall tumor volumes of the brequinar-treated mice were lower than those of control mice (Fig 3B and 3D, and S5B and S5C Fig). Similar to the tumor volumes, tumor weights (measured at the endpoint) were also lower in the brequinar-treated mice (Fig 3C and 3D and S5B Fig). In contrast, the total body weights of the mice treated with 10 mg/kg brequinar daily for 55 days were unaffected (Fig 3E and S3C Fig), which suggests that brequinar does not cause overt toxicity in other tissues with this dose and schedule.

Interestingly, the control group tumors were highly vascularized, whereas the brequinar-treated tumors were paler (Fig 3F, left panels and S5B Fig). In fact, the mRNA levels of the human *vascular endothelial growth factor A* (*VEGFA*), known to increase vascularity in glioblastoma [57], were lower in the brequinar-treated group after 55 days of treatment (Fig 3F, right panel). Nonetheless, HIF1α, a transcription factor known to induce the expression of *VEGFA* in hypoxia [58], was not detectable in either the control or brequinar-treated tumors (Fig 3G). It is possible that *VEGFA* expression is regulated by other mechanisms in the LN229 xenograft tumors. Moreover, in agreement with our results *in vitro* (Fig 2F, 2G, 2K and 2M–2O), the tumors treated with brequinar had higher expression of p53 as well as the neuronal differentiation marker acetyl-tubulin (Fig 3G), which is supported by recent work showing that inhibition of the *de novo* pyrimidine synthesis pathway increases the differentiation of glioblastoma tumor cells [34, 35].

## Inhibition of DHODH with brequinar reduces pyrimidine levels in the LN229 tumor xenografts more efficiently than in normal tissues

To confirm that brequinar inhibited the *de novo* pyrimidine biosynthesis *in vivo* in the tumor xenografts, we used LC-MS/MS to measure the amounts of brequinar as well as UMP, UDP, UTP, and uridine present in tumor tissue in the control and brequinar-treated group. As expected, brequinar was only present in the brequinar-treated animals, and UMP, UDP, UTP and uridine were dramatically lower in the brequinar-treated group (Fig 4A and S1 Table), confirming that brequinar inhibited the *de novo* pyrimidine biosynthesis in the tumor xenografts. Given that tumor growth and the amount of pyrimidine metabolites were reduced in the brequinar-treated mice, we quantified the 47S pre-rRNA and the

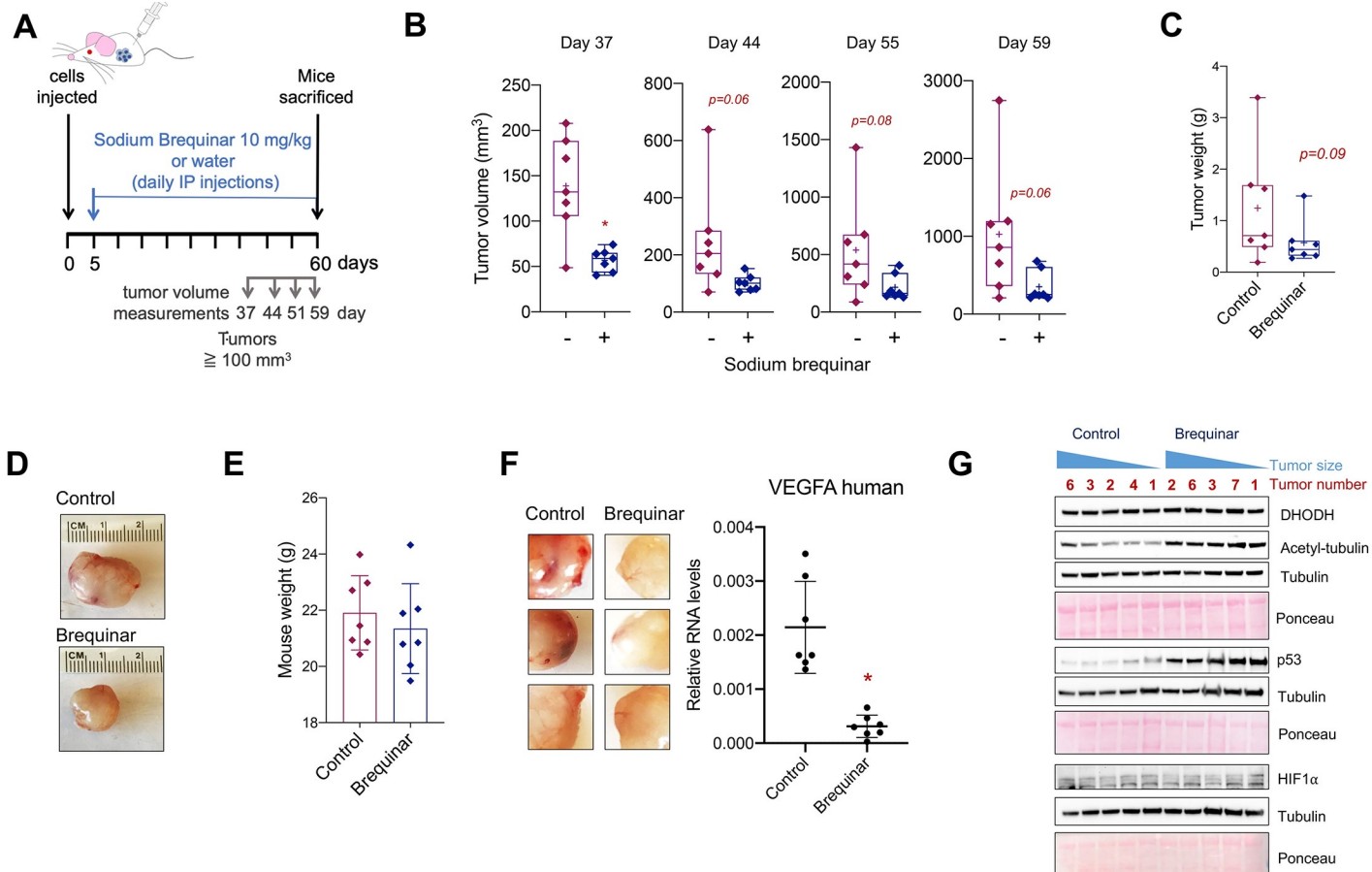

**Fig 3. Blocking DHODH with brequinar reduced reduces glioblastoma tumor xenografts growth *in vivo*.** (A) Representation of the subcutaneous xenograft experiment using LN229 cells. Mice were treated with 10 mg/kg brequinar with daily intraperitoneal injections (IP). (B) Tumor volume measurements of LN229 xenografts once the tumors reached 100 mm³. (C) Tumor weight of LN229 xenografts at day 60 (end of experiment). See also Supplementary S5B and S5C Fig. (D) Representative xenograft tumors from control and brequinar-treated mice. See also Supplementary S5B Fig. (E) Mouse weights at day 60 before the tumors were harvested. (F) Three representative control and brequinar-treated LN229 xenografts tumors showing increased blood vascularity in the control group (left panel) and qPCR for *VEGFA* in the LN229 xenografts. (G) Western blot of 5 control and brequinar-treated LN229 xenograft tumors for DHODH, p53, acetyl-tubulin, and HIF1α. * indicates p-values ≦ 0.05. Numerical values for each of the experiments represented are available in S3 Data.

28S and 18S rRNAs as well as *ACTIN* in the tumor xenografts. Although the 47S pre-rRNA did not change, the brequinar-treated tumor xenografts had lower levels of 28S and 18S rRNAs (Fig 4B). This explains the decrease in tumor growth of the brequinar-treated xenografts. *ACTIN* mRNA levels were also slightly lower in the brequinar group (S5D Fig), which is likely due to the prolonged treatment with brequinar in this experiment as opposed to the 24 h treatment in the *in vitro* experiments (Fig 1I and 1J). We then compared the amounts of 47S pre-rRNA, 28S and 18S rRNAs and *ACTIN* levels with each tumor weight to see whether a larger tumor size correlated with higher levels of rRNAs. Surprisingly, the levels of 47S pre-rRNA positively correlated with tumor size only in the control group and not in the brequinar-treated group (S5E Fig).

To determine whether brequinar treatment affects pyrimidine synthesis in other non-transformed differentiated tissues in the mice, we collected brain and liver tissues in addition to serum from the same mice used for the xenograft experiment. LC-MS/MS was used to quantify brequinar, UMP, UDP, UTP, and uridine in all harvested tissues. As expected,

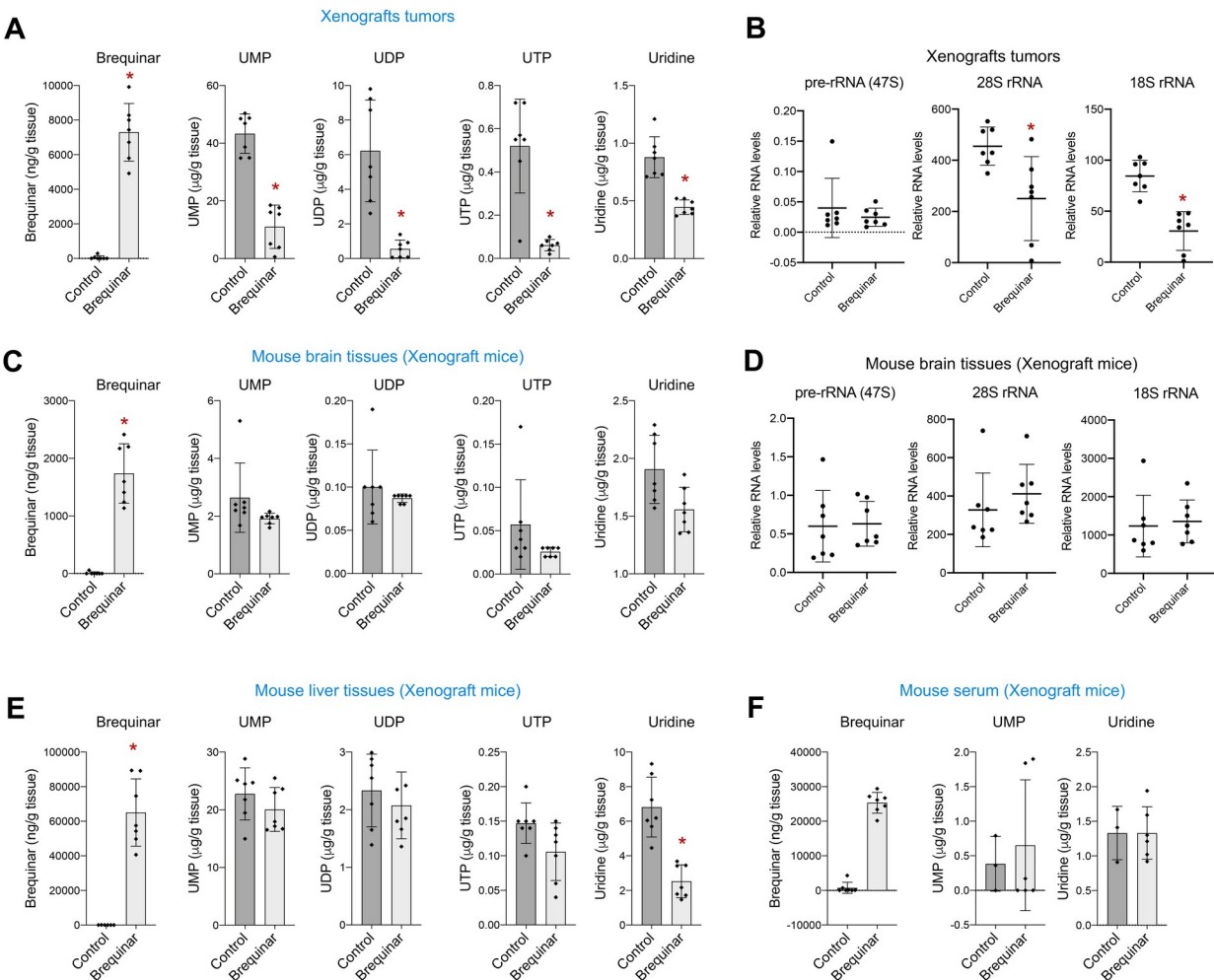

**Fig 4. Blocking DHODH with brequinar reduces the production of ribosomal RNA in glioblastoma tumors xenograft *in vivo*.** (A) Amounts of brequinar, UMP, UDP, UTP and uridine in the LN229 xenografts measured by LC-MS/MS. (B) qPCR of 47S pre-rRNA and 28S and 18S rRNAs in the LN229 xenografts. 47S pre-rRNA, 28S and 18S levels were normalized by *ACTIN* mRNA levels. (C) Amounts of brequinar, UMP, UDP, UTP and uridine measured by LC-MS/MS in the brain tissues of mice used for xenografts. (D) qPCR of 47S pre-rRNA and 28S and 18S rRNAs in the brain tissues of the mice used for LN229 xenografts. 47S pre-rRNA, 28S and 18S levels were normalized by *ACTIN* mRNA levels. (E) Amounts of brequinar, UMP, UDP, UTP and uridine measured by LC-MS/MS in the liver tissues of mice used for xenografts. (F) Amounts of brequinar, UMP, and uridine measured by LC-MS/MS in the serum of mice used for xenografts. Last brequinar injection was 3 h before harvesting the tissues. Numerical values for each of the experiments represented are available in S4 Data.

brequinar was only present in the brain, liver, and serum of the treated mice. UMP, UDP, UTP, and uridine levels in the brain tissues were not affected by the treatment with brequinar (Fig 4C and S1 Table). Consistently, none of the rRNA forms nor *ACTIN* were reduced in the brain tissues of mice treated with brequinar (Fig 4D and S5F Fig). Similarly, in the liver tissues of brequinar-treated mice only uridine but not UMP, UDP or UTP, was significantly decreased (Fig 4E). It is possible that uridine is used to synthesize pyrimidines by the salvage pathway (Fig 1C) in this differentiated tissue. Moreover, UMP and uridine levels in the serum (UDP and UTP were undetectable) of brequinar-treated mice were also unaffected (Fig 4F). Together these results demonstrate that the effects of brequinar in limiting pyrimidine production are more specific to glioblastoma cells and thus may be a safe drug with minimal side effects.

## *De novo* pyrimidine biosynthesis regulates nucleolar activity in glioblastoma cells

Active rDNA transcription governs the overall organization of the nucleolus [59]. Indeed, the morphology of the nucleolus reflects its functional state. The nucleolus has 3 subcompartments in which different steps of ribosome production occurs: rDNA is transcribed in the fibrillar center (FC); pre-rRNA processing takes place in the dense fibrillar center (DFC); and the assembly of ribosome subunits happens in the granular component (GC) [23]. When rDNA transcription is inhibited, the nucleolus undergoes nucleolar stress [60], which is characterized by the redistribution of nucleolar proteins to its periphery, creating a rounder shape and condensed appearance. The presence of nucleolar stress can be visualized by following the distribution of specific markers that localize in the different compartments of the nucleolus [60].

Because glioblastoma cells relied on the *de novo* pyrimidine biosynthesis to generate nucleotides and support high rates of rDNA transcription to produce rRNA, we asked whether the inhibition of the *de novo* biosynthesis led to nucleolar stress. Thus, we analyze the nucleolar morphology upon brequinar treatment by evaluating the distribution of proteins that localize in the different compartments of the nucleolus such as UBF, which localizes in the FC, and NPM1, which localizes in the GC. Brequinar treatment led to nucleolar stress as seen by UBF (Fig 5A and S6A Fig) and NPM1 (Fig 5B and S6C–S6E Fig) redistribution to the periphery of the nucleolus. This effect was reversed by the addition of uridine. Uridine did not cause apparent effects in the nucleolar morphology in the control conditions (Fig 5A and 5B and S6A–S6E Fig). Importantly, the inhibition of the *de novo* pyrimidine biosynthesis with brequinar did not alter UBF distribution or the nucleolar morphology of the non-transformed ARPE cells (S6B Fig), indicating that only the transformed glioblastoma cells suffer nucleolar stress upon DHODH inhibition.

Nucleolar stress leads to the stabilization of p53 through the release of ribonucleoproteins that block the interaction of p53 with MDM2, an E3 ubiquitin ligase that tags p53 for proteasomal degradation [60–62]. Indeed, we found by Western blotting that brequinar treatment led to increased p53 protein (Fig 2F, 2G, 2K and 2M–2O, and S2F, S2G and S2J). Accordingly, p53 IF indicated that nuclear p53 increased in the glioblastoma cells treated with brequinar and that the addition of uridine rescued this effect (Fig 5C and 5D). Our results are in agreement with a recently published study showing that inhibition of DHODH in colon and mammary cancer cells leads to decrease 47S pre-rRNA abundance and accumulation of p53 [63].

In parallel to demonstrating that brequinar caused nucleolar stress leading to p53 accumulation, we confirmed by IF with an anti-total rRNA antibody (Y10b) that the abundance of rRNA at the single-cell level was also reduced upon DHODH inhibition. This antibody detects the 47S pre-rRNA as well as the mature rRNA. Confirming our biochemical experiments, our IF results indicate that brequinar decreases the abundance of total rRNA both in the nucleolus and cytoplasm, and that uridine rescues this effect (Fig 5E, 5F, and S6F and S6G Fig). To determine whether the decrease in rRNA production caused by inhibition of the *de novo* pyrimidine biosynthesis affected the assembly of ribosome subunits, we performed polysome profiling of LN229 treated with brequinar for 72 h in the presence or absence or uridine. Cells treated with brequinar had fewer monosomes in comparison with the 40S and 60S subunits fractions (Fig 5G). The presence of uridine rescued this effect as seen by an increased number of monosomes. Interestingly, uridine in the control condition also led to increase number of 80S monosomes, which suggests that additional uridine promotes the production of rRNA. In summary, our data show that the inhibition of the *de novo* pyrimidine biosynthesis by targeting the enzymatic activity of DHODH leads to efficient inhibition of rRNA production and nucleolar stress and therefore, decreases proliferation specifically in glioblastoma cells.

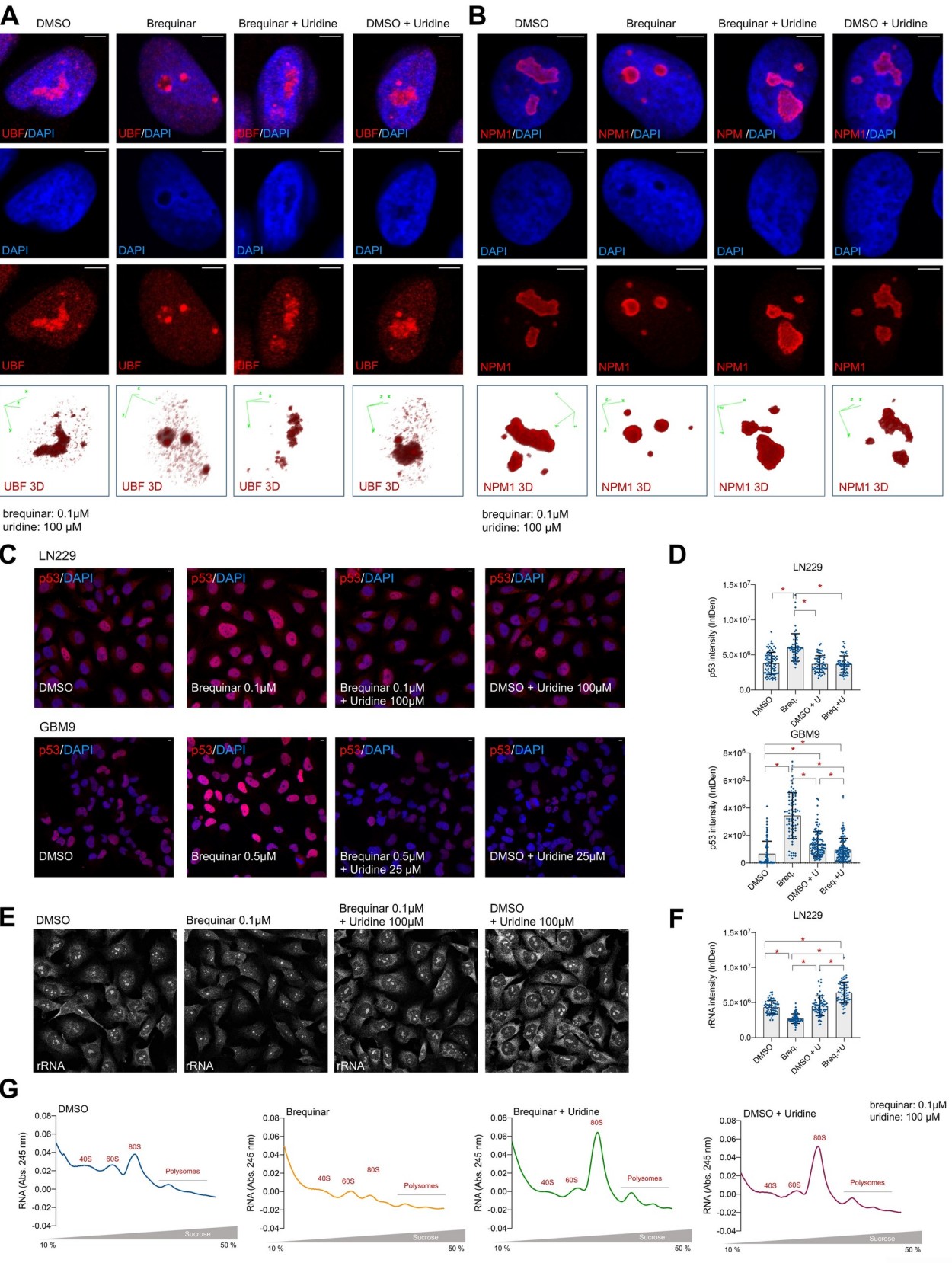

**Fig 5. The inhibition of *de novo* pyrimidine biosynthesis causes nucleolar stress in glioblastoma cells.** (A) IF of UBF in LN229 with or without brequinar and uridine for 24 h. See also Supplementary S4A Fig. (B) IF of NPM1 in LN229 cells with or without brequinar and uridine for 24 h. See also Supplementary S4C Fig. Lower panels in (A) and (B) show UBF and NPM1 3D representation from Z-stack images. (C) IF of p53 in LN229 (upper panels) and GBM9 (lower panels) in the presence of brequinar with or without uridine for 24 h. (D) Quantification of (C) with Image J. (E) IF of rRNA in LN229 cells with the anti-rRNA Y10b antibody with or without brequinar and uridine for 24 h. See also Supplementary S4F Fig. (F) Quantification of (E) with Image J. (G) Ribosome profiling of LN229 treated with or without brequinar and uridine for 72 h. This experiment was done twice with similar results. For all experiments, media with drugs and metabolites were replaced the day after seeding, and cells harvested at the indicated time points. Scale bars = 5 μm. Numerical values for each of the experiments represented are available in S5 Data.

## Methods

### Cell proliferation and cell cycle analysis

All cells were cultured in DMEM with 10% FBS and 100 U/mL penicillin/streptomycin.

For cell proliferation, six-well or 12-well plates were seeded with $1.5 \times 10^5$ or 50,000 cells, respectively. The day after seeding, the media were replaced with fresh media containing the appropriate inhibitors and metabolites. Media with drugs and metabolites were replaced every 2 days for 6 days for the experiments shown in Fig 2A, and 2B and S2A, S2B and S2C Fig. For the experiments shown in Fig 2D, 2E, 2I and 2J and S2D, and S2E Fig, media with inhibitors and metabolites were replaced the day after seeding, and cell proliferation was measured 4 days later. Cell proliferation was measured by crystal violet staining and 595 nm absorbance quantification of the solubilized die. Values are represented as relative growth rate after normalizing by the control condition. Experiments were performed a minimum of 3 times. For cell cycle analyses, two million cells were seeded. The day after seeding, fresh media with drugs and uridine was added. After 24 h of treatment, cells were trypsinized and harvested and washed with PBS. Then, cells were resuspended in 0.5 mL PBS and fixed by adding 70% cold ethanol in drop-wise manner on a vortex. Cells were at -20°C for at least 2 h before staining. Following fixation, cells were wash twice with 5 mL of PBS containing 2% BSA, and 1 mM EDTA. Cells were stained in 500 μL of PI solution (50 μg/mL propidium iodide, 100 μg/mL RNAse A, and 2 mM $MgCl_2$) for 20 min at 37 °C. Cell cycle was analyzed using FlowJo software. Cell cycle analysis were performed three times with similar results.

### TCGA and CGGA data analysis

RNA expression data for *CAD*, *DHODH*, and *UMPS* in grade II to IV (GBM) gliomas was obtained from the RNA-seq experiments deposited in The Cancer Genome Atlas (TCGA) Program through the Pancancer TCGA project (https://portal.gdc.cancer.gov) and the Chinese Glioma Genome Atlas (http://www.cgga.org.cn). TCGA-GBM and TCGA-LGG FPKM gene expression and clinical data were downloaded using the R package 'TCGABiolinks' [64].

### Transient transfections

siRNA transfections were performed using Lipofectamine RNAiMAX (Invitrogen). Briefly, 150,000 cells were seeded and transfected with 3 μL siRNA (20 μM) and 3 μL Lipofectamine previously mixed on OPTI-MEM for 15 min. siRNAs used in this study are listed in S2 Table. The day after transfection, the media was replaced with fresh media with or without uridine as indicated in the text.

### RT-qPCR

Total RNA was extracted with TRI Reagent solution (Sigma) following the manufacturer's instructions, and cDNA was produced with the iScript RT Supermix for RT-qPCR (BIO-RAD). RNA levels were measured by quantitative PCR with the iTaq Universal SYBR

Green Supermix (BIO-RAD). For the experiments *in vitro*, expression levels for 47S pre-rRNA, 18S rRNA and 28S rRNA were normalized to β-actin expression. β-actin levels were normalized to RNA load for the cDNA production. Relative RNA levels are shown as a ratio between the experimental and control conditions. For the *in vivo* experiments, expression levels for 47S pre-rRNA, 18S rRNA, 28S rRNA and *VEGFA* are shown for each tumor after normalizing the expression levels of each locus to β-actin mRNA levels. β-actin levels were normalized to RNA load for the cDNA production. The primers used in this study are listed in S2 Table.

## Western blot

Cells were harvested and lysed in RIPA buffer (25 mM Tris-HCl pH 7.4, 150 mM NaCl, 1% NP-40, 0.5% sodium deoxycholate, 0.1% SDS + protease and phosphatase inhibitors and MG132). Lysates were set on ice for 15 min and resuspended every 5 min. Then, cell lysates were sonicated on high intensity using 30-sec ON/OFF cycles for 5 min. Cell debris was centrifuged at maximum speed for 15 min. Supernatants were collected and kept at -80 °C until use. For protein electrophoresis, samples were mixed and boiled for 5 min with denaturant buffer Laemli at 1X final concentration. Samples were then run on a 4%-12% gradient acrylamide gel and transfer to a nitrocellulose membrane. Membranes were blocked with 5% BSA for at least 1 h at room temperature (RT) and incubated overnight with primary antibodies dissolved in TBS-T 5% BSA. Three 10-min TBS-T washes were performed before adding secondary antibodies dissolved in TBS-T 5% BSA for 1 h at RT. Membranes were washed 3 times with TBS-T for 10 min and imaged using BIO-RAD chemiluminescence imager. The antibodies used in this study are indicated in S2 Table. Western blot quantifications were performed with Image J, densitometry was obtained for each of the proteins measured. Then each quantification was normalized by the amount of loading control (actin or tubulin). Next, a ratio was obtained normalizing to the DMSO condition.

## Immunofluorescence

Cells were grown on glass coverslips and fixed with 4% paraformaldehyde PBS for 15 min, permeabilized with 0.5% Triton X-100 for 20 min and blocked with 5% BSA in PBS for 30 min to 1 h. Primary antibodies were diluted at 1:1000 in 5% BSA-PBS and incubated overnight at 4˚C. The cells were washed 3 times for 10 min in PBS and incubated with secondary antibody. Secondary antibodies (Alexa Fluor 594 Goat Anti-Mouse IgG H+L and Alexa Fluor 594 Donkey Anti-Goat IgG H+L -Thermo Fisher), were diluted at 1:1000 in 5% BSA-PBS, incubated for 1 h at RT, washed with PBS 3 times for 10 min and mounted with Mowiol mounting media. The second PBS wash contained DAPI at 5 μg/mL to stain the nuclei. Cells were imaged with a Zeiss LSM780 inverted confocal microscope. All antibodies used in this study are indicated in S2 Table. Intensity quantification were performed using Image J using integrated density quantification for individual cells.

## Xenografts

For xenograft experiments in S5A Fig, 1 x 10⁶ LN229 cells were injected into the flank of 8-week-old female NOD/SCID mice (n = 5 per group). On day 6, sodium brequinar was administered by intraperitoneal (IP) injections every 3 days at 15 mg/kg over a period of 60 days. At the end of the experiment, tumors were harvested and weighed. For the experiment shown in Fig 3A, 3.5 x 10⁶ LN229 cells were injected into the flank of 8-week-old female NOD/SCID mice (n = 7 per group, Day 0). On day 5, sodium brequinar was administered by daily IP at 10 mg/kg for 55 more days. Tumor volumes were recorded once they reached 100

mm$^3$ at day 37, 44, 51, and 59. At the end of the experiment the mice were weighed, and the tumors were harvested and weighed. Additionally, serum and brain tissues were collected from each mouse. For serum purification, mouse blood was collected, placed on ice at least 20 min and centrifuged at 14,000 rpm. Then, the supernatant was collected as the serum fraction. Tumor xenografts and normal brain tissues were snap frozen using liquid nitrogen for use in the LC-MS/MS analysis as described below or for RNA and protein extraction. All animal experiments were performed following UTSW IACUC guidelines (APN# 2017–101798).

## LC-MS/MS

All mice samples for LC/MS measurements were harvested 3 h after the last brequinar injection. Tissues (xenograft tumors, brain, and liver tissues) were weighed and homogenized in a 3-fold volume of PBS (3 x weight of tumor in g = vol PBS in mL; total homogenate volume (in mL) = 4 X weight of tissue). Cells from *in vitro* experiments were harvested and lysed with 80% cold methanol to extract the cell metabolites. The samples were snap-frozen three times in liquid nitrogen. Next, lysates were centrifuged at 15,000 rpm for 15 min and the supernatant were collected in a new tube. Supernatants were dried down by speedvac overnight. For nucleotide measurements, serum and tissue aliquots (xenograft tumors or brain tissues) were crashed with a 4X volume of MeOH (80% MeOH final), vortexed, incubated for 10 min at RT, and spun 5 min at 16,100 x g at 4˚C to precipitate protein. The supernatants were transferred to new tubes and dried down by speedvac. The protein pellets were saved and frozen. The samples were resuspended in Buffer A containing the internal standards (IS) UMP-$^{13}$C$_9$,$^{15}$N$_2$ (UMP-IS) and tolbutamide (Resuspension Solution). The materials were dissolved by vortexing, incubating for 10 min at RT, vortexing again, incubating for 10 min at 37˚C, and vortexing a final time. The samples were spun at 16,100 x g for 5 min and the supernatants were kept. To generate nucleotide standards curves, supernatants from control tumor, serum and brain samples were diluted 1:5000 in Resuspension Solution and used for the nucleotide standard curves. A 1:25000 dilution of control samples was used for the uridine standard curve. Various known concentrations of UMP, UDP, UTP, and uridine were spiked into the Resuspension Solution containing diluted control matrix to create each standard curve. UTP, UDP and UMP were detected by LC-MS/MS using a Sciex QTRAP 6500+ mass spectrometer, operating in MRM (multiple reaction monitoring) mode, coupled to a Shimadzu (Columbia, MD) Prominence LC. The following precursor to fragment ion transitions were optimized for each nucleotide: UMP 325.084 to 97.1; UDP 405.053 to 97.0; UTP 484.984 to 96.9; UMP-IS 336.092 to 102.0. A Thermo Scientific (Waltham, MA) Biobasic Anion Exchange (AX) column (2.1 x 50 mm, 5 micron packing) was used for chromatography with the following conditions: Buffer A: 7:3 water:acetonitrile 10 mM NH$_4$ acetate, pH 6; Buffer B: 7:3 water: acetonitrile 1 mM MH$_4$ acetate, pH 10.5; flow rate 0.5 mL/min; 0–1 min 0%B, 1–2.5 min 35%B, 2.5–5 min 35%B, 5–7 min 65%B, 7–10 min 65%B, 10–10.5 min 100%B, 10.5–15 min 100%B, 15–15.5 min 0%B, 15.5–20.5min 0%B. Uridine was also detected by LC-MS/MS (uridine transition 243.0 to 109.9; tolbutamide IS 269.1 to 169.9) using a Sciex 4000 QTRAP operating in MRM mode and coupled to a Shimadzu Prominence LC. A Phenomenex (Torrance, CA) *Synergi* Polar-RP column (2.0 X 150mm, 4 micron packing) was used for chromatography with the following conditions: Buffer A: ddH$_2$O with 0.2% acetic acid. Buffer B: acetonitrile with 0.2% acetic acid, flow rate 0.4 mL/min; 0–0.5 min 5% B, 0.5–1.5 min gradient to 95% B, 1.5–3.5 min 95% B, 3.5–3.6 min gradient to 5%B, 3.6–5.5 min 5% B.

For brequinar measurements, aliquots of serum or tissue homogenates were mixed with a 3X volume of acetonitrile containing formic acid and the tolbutamide IS, vortexed for 15 seconds, incubated at RT for 10 min, and spun twice in a tabletop, chilled centrifuge for 5 min at

16,100 x g. The supernatants were analyzed by LC-MS/MS. To generate a brequinar standard curve, varying concentrations of brequinar standards were spiked into control tumor homogenate, blank commercial serum, or blank brain homogenate and processed like samples. Brequinar detected by LC-MS/MS (transition 375.897 to 133) using a Sciex 4000 QTRAP operating in MRM mode and coupled to a Shimadzu Prominence LC. An Agilent (Santa Clara, CA) C18 XDB column (5 micron packing 50 X 4.6 mm size) was used for chromatography with the following conditions: Buffer A: Water + 0.1% formic acid; Buffer B: acetonitrile + 0.1% formic acid; flow rate 1.5 mL/min; column; 0–1.5 min 20% B, 1.5–2.0 min gradient to 100% B, 2.0–4.5 min 100% B, 4.5–4.6 min gradient to 20% B, 4.6–5.5 20% B. Data were analyzed using GraphPad for nucleotide quantification. The blank was subtracted from the standard curve and the subtracted values were used to determine analytical concentration of the compounds. For brequinar quantification, the Analyst software Sciex MS computer was used to calculate the compound concentrations.

## Ribosome profiling

4–5 million cells were harvested at 60%-80% confluency and resuspended in 500 μL lysis buffer (20 mM Tris pH 7.4, 5 mM MgCl$_2$, 100 mM NaCl in DPEC-treated dH$_2$O + 100 μg/mL CHX + Protease inhibitor cocktail (1/100 from stock) + 0.1% NP-40). The lysates were passed through a 20G syringe and incubated on ice for 15 min with resuspending every 5 min. Samples were then centrifuged 12,000g 4˚C for 10 min to pellet nuclei and mitochondria. The supernatants were collected. RNA amounts were quantified by nanodrop, and all samples were set at the same RNA concentration to load same amount of RNA in the sucrose gradient columns. Gradients were made by solubilizing different sucrose amounts in buffer 0% (20 mM Tris pH 7.4, 5 mM MgCl$_2$, 100 mM NaCl in DPEC-treated dH$_2$O + 100 μg/mL CHX). The sucrose gradient column comprises 2 mL of 10%, 20%, 30%, 40%, and 50% sucrose solutions. Starting the gradient column by the most concentrated sucrose solution. Each time a different gradient solution was added, the solution was frozen at -80 $^o$C for at least 20 min before applying the next sucrose % solution. The columns were kept at -80 $^o$C until use. Before use, the gradient column tubes were thawed at 4 $^o$C overnight to form continuous sucrose gradient. Lysates were loaded on vertical gradient columns on ice. The column walls were cleaned to remove any liquid before balancing, weighed and balanced with <0.005 g difference of weight by adding buffer 0%. Balanced columns were centrifuged in a swinging bucket rotor at 34,000 rpm for 2 h at 4˚C, acc = 8, dec = 0. The BioLogic LP low-pressure chromatography system (BIO-RAD) was used to analyze the centrifuged fractions. The samples were run at 1 mL/min, and the UV recorded. The UV intensities were plotted as arbitrary units (A.U.) in a XY graph. This experiment was performed twice with similar results.

## Inhibitors and metabolites

Drugs for *in vitro* experiments: brequinar (24445, Cayman Chemical) dissolved in DMSO. ML390 (21395 Cayman Chemical) dissolved in DMSO. 5-Fluorouracil (F6627, Sigma) dissolved in DMSO. Temozolomide (T2577, Sigma) dissolved in DMSO. Equivalent volume of DMSO was used as control conditions. Nucleotides for *in vitro* experiments: uridine (AAA1522706, Alfa Aesar) dissolved in water. Equivalent volume of water was used as control conditions.

Analytical Standards for LC-MS/MS analysis: uridine (Sigma U3750), UMP (Sigma U6375), UDP (Sigma 94330), UTP (Sigma U6625) and UMP-$^{13}$C$_9$,$^{15}$N$_2$ (Sigma/Isotech 651370) were all dissolved in water. Tolbutamide (Sigma T0891) and Brequinar (24445, Cayman Chemical) were dissolved in DMSO. Drugs for *in vivo* (subcutaneous xenografts)

experiments: brequinar sodium (Tocris 6196/50) dissolved in water. Equivalent volume of autoclaved water was used as control conditions.

## Quantification and statistical analysis

All statistical analyses were performed using two-tailed unpaired T-student statistical analysis, p ≦0.05 was considered statistically significant. All values are reported as mean ± SD in each figure.

## Discussion

Here, we show that the activation of the *de novo* biosynthesis of pyrimidines is an adaptive mechanism used by glioblastoma cells to sustain the high transcriptional rates exhibited by cancer cells. As others previously demonstrated, DHODH inhibition leads to a decrease in proliferation of glioblastoma cells [34, 35]. While previous studies attributed this phenotype to a decrease in stemness of glioblastoma-initiating cells [34, 35], our current results indicate that the decrease of glioblastoma cell proliferation upon inhibition of DHODH is caused by a specific decrease in pre-rRNA and rRNA abundance, which leads to nucleolar stress and insufficient protein synthesis (Fig 6). While inhibition of DHODH causes a decrease in rRNA production in glioblastoma cells, it does not affect *ACTIN* abundance at short incubation periods. It is possible that longer treatments or higher amounts of the DHODH inhibitors brequinar or ML390 would affect transcription more generally. In addition, our xenograft data indicate that the inhibition of DHODH did not affect the weight of the mice or the production of pyrimidines and rRNA in normal brain, and liver tissues. The fact that non-cancer cells do not present such high rates of rRNA production could explain the low toxicity observed in the normal tissue of the mice. It is likely that normal tissues do not depend on *de novo* biosynthesis of pyrimidines to maintain the intracellular pyrimidine pool, and that the production of pyrimidine by the salvage pathway is sufficient to fulfill their pyrimidine demands when the *de novo* pyrimidine biosynthesis is inhibited in normal tissues. This suggests that inhibition of DHODH specifically impairs the production of pyrimidines in tumor tissues with low side effects. Consequently, the impairment of the pre-rRNA synthesis upon DHODH inhibition leads to nucleolar stress specifically in glioblastoma cells but not in non-transformed cells. Our results on rRNA and nucleolar stress are in agreement with previous studies showing that *de novo* purine biosynthesis is necessary for rRNA synthesis and proliferation in glioblastoma [36] and that DHODH inhibition with leflunomide leads to decreased rRNA synthesis and p53 stabilization in mammary and colon cells [63]. We propose that glioblastoma cells rely more heavily on the *de novo* biosynthesis of nucleotides than in the salvage pathways to sustain rRNA production and proliferation, and therefore our work highlights the metabolic vulnerabilities of glioblastoma tumors.

The poor prognosis associated with glioblastoma warrants major efforts towards improving current therapeutic interventions. Currently, the most common regimen of treatment for glioblastoma patients is surgical resection followed by radiotherapy and chemotherapy with TMZ [29]. Frequently, glioblastoma patients develop chemotherapy resistance mainly due to increase expression of MGMT or other DNA repair enzymes [29, 30]. In fact, hypermethylation of the *MGMT* promoter, which leads to decreased *MGMT* expression correlates with long-term survival of glioblastoma patients [29, 30, 65, 66]. Interestingly, brequinar treatment decreases the expression of MGMT in cells where MGMT is highly expressed and thus are more likely to become resistant to TMZ. It is possible that the *MGMT* locus is highly transcriptionally active as a mechanism to increase MGMT levels and DNA repair. The decrease in pyrimidine levels caused by brequinar, could then decrease the levels of *MGMT* mRNA and

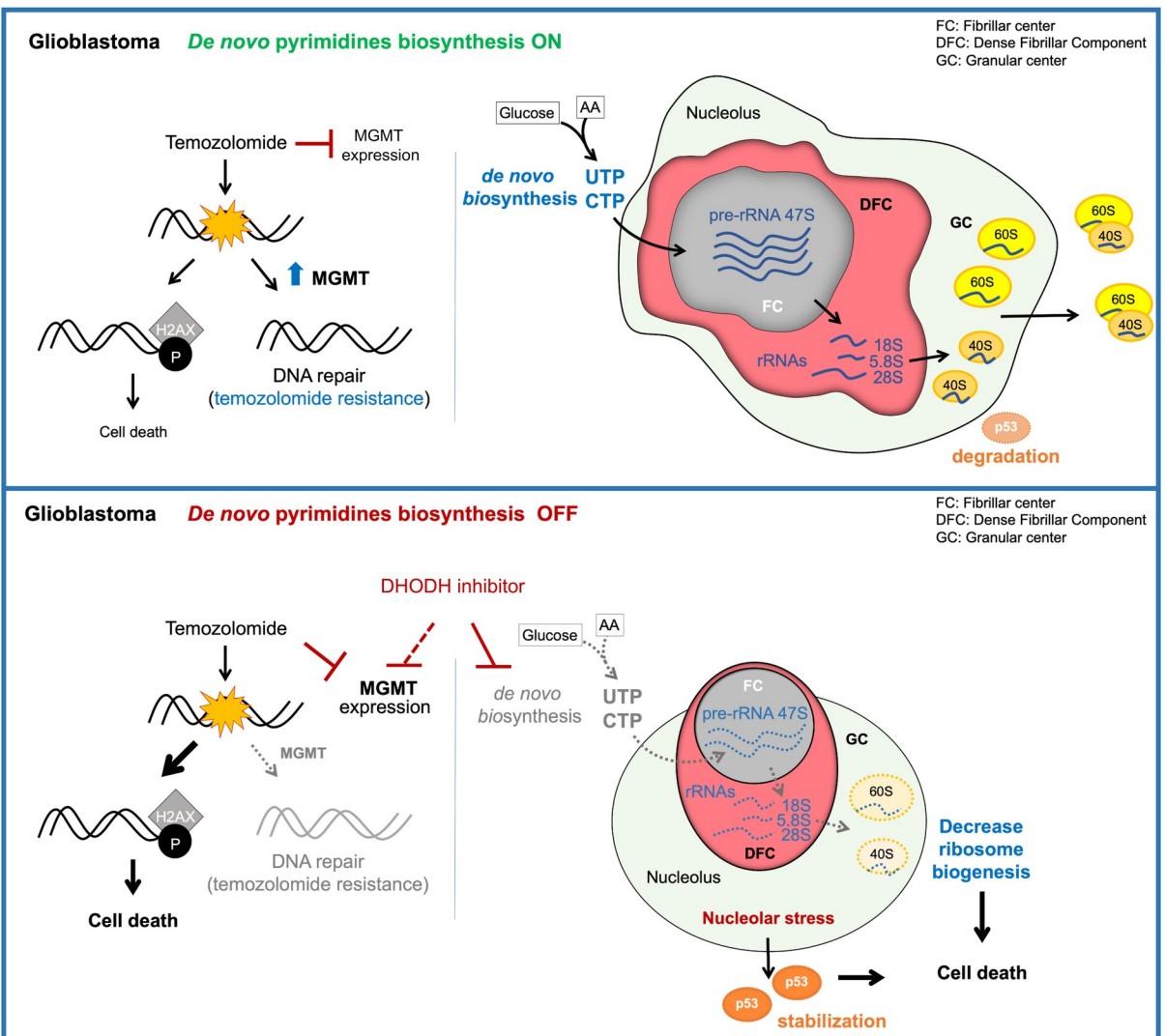

**Fig 6. Inhibition of *de novo* pyrimidine biosynthesis reduces ribosomal RNA production and the ability to repair DNA in glioblastoma.**
Inhibition of the *de novo* pyrimidine biosynthesis by blocking the activity of DHODH leads to reduced pyrimidine nucleotide availability and thus affects the synthesis of pre-rRNA and rRNA. This in turns induces nucleolar stress leading to changes in nucleolar morphology, the stabilization of p53, cell cycle arrest and cell death. Furthermore, brequinar and TMZ treatment leads to decreased expression of the DNA damage repair enzyme MGMT. The decrease in MGMT levels leads to an increase in H2AX phosphorylation and thus enhances the sensitivity of glioblastoma cells to DNA damage-induced death.

therefore decrease the repair of DNA damage caused by TMZ. Alternatively, it is possible that brequinar induces changes in the methylation status of the *MGMT* promoter or that other unknown pre- or post-transcriptional regulatory mechanisms are responsible for the decrease in MGMT protein levels upon brequinar treatment. However, further experiments are needed to address this question. Nonetheless, here, we show that glioblastoma TMZ-resistant cells are still dependent on functional DHODH to proliferate, suggesting that targeting DHODH can provide a unique therapeutic intervention route to treat TMZ-resistant tumors. Our results indicate that the combination of TMZ and brequinar in glioblastoma TMZ-resistant cells does not lead to synergistic or strongly additive effects. This suggest that the inhibition of DHODH can be used as a sequential step after TMZ treatment and not necessarily in combination with

it, indicating that it could become an alternative approach to treat glioblastoma patients that develop TMZ resistance. Overall, our findings suggest new therapeutic possibilities for glioblastoma patients by blocking the production of ribosomes through the specific inhibition of the *de novo* pyrimidine biosynthesis pathway.

## Supporting information

**S1 Fig. *De novo* pyrimidine biosynthesis pathway maintains efficient ribosomal RNA synthesis in glioblastoma.** (A) mRNA expression of lower grade (II-III) to higher grade (IV/Glioblastoma) gliomas from patients whose tumors are archived in the CGGA database. (B) Table showing the characteristics of the cell lines used in the present manuscript. (C) Brequinar levels quantified by LC-MS/MS in LN229 cells treated with 0.1 μM brequinar. Numerical values for each of the experiments represented are available in S6 Data.
(PDF)

**S2 Fig. Inhibition of the *de novo* pyrimidine biosynthesis pathway decreases proliferation of glioblastoma cells in temozolomide-sensitive and -resistant cells.** (A) Relative proliferation of non-transformed retinal epithelial ARPE and the glioblastoma cells LN229, GBM9 and SF188 in the presence of increasing amounts of the DHODH inhibitor, ML390. Media with drugs was replaced every 2 days for 6 days. Proliferation was assessed by crystal violet staining. (B) Relative proliferation of ARPE cells and the LN229, GBM9 and SF188 GBM cells in the presence of ML390. Media with drugs was replaced every 2 days for 6 days. N = 2–5. (C) Relative proliferation of LN229, GBM9 and SF188 cells in increasing amounts of uridine. Media with uridine was added the day after seeding. Media and uridine were replaced every 2 days, and proliferation was 6 days after treatment. N = 4. (D) Relative proliferation of ARPE, LN229 and GBM9 cells with or without brequinar and with or without uridine. Media with drugs and uridine were replaced the day after cell seeding, and proliferation assessed after 4 days. N = 3. (E) Relative proliferation of the LN229 and GBM9 cells in the presence of increasing amounts of temozolomide (TMZ). Media with drug were replaced the day after seeding, and proliferation assessed after 4 days. N = 4–5. (F) Western blot quantification by Image J of Fig 2F and additional experimental replicates. (G) Western blot quantification by Image J of Fig 2G and additional experimental replicates. N = 2. (H) Relative proliferation of SF188 TMZ-sensitive or -resistant cells with or without TMZ, brequinar or brequinar + TMZ with or without uridine normalize to SF188 TMZ-sensitive DMSO condition. Also see Fig 2J. (I) Western blot quantification by Image J of γH2AX in Fig 2K and additional experimental replicates. The results are represented for each experimental replicate. N = 2. (J) Western blot quantification by Image J of p53 and MGMT in Fig 2K and additional experimental replicates. N = 2 for p53, N = 3 for MGMT. (K) Cell cycle representation of results in Fig 2L. Asterisks indicate p-values ≦ 0.05. Numerical values for each of the experiments represented are available in S7 Data.
(PDF)

**S3 Fig. DHODH knockdown reduces proliferation but does not decrease pre-rRNA synthesis or pyrimidine abundance in glioblastoma cells.** (A) Relative proliferation of GBM9 cells upon *DHODH* knockdown with or without uridine. Proliferation was assessed by crystal violet staining after 4 days of siRNA transfection. Fresh media with uridine were added the day after siRNA transfection. Uridine did not rescue the effects of *DHODH* knockdown in proliferation. Western blot for DHODH on the right panel. N = 3. (B) Relative proliferation of LN229 cells upon *DHODH* knockdown with or without uridine. Proliferation was assessed by crystal violet staining after 4 days of siRNA transfection. Fresh media with uridine were added the day after siRNA transfection. Uridine did not rescue the effects of *DHODH* knockdown in

proliferation. N = 3. (C) qPCR of *DHODH*, 47S pre-rRNA and *ACTIN* in LN229 after 3 days of transfection with 2 different siRNA for *DHODH*. N = 2–6. (D) Western blot of DHODH in LN229 after 3 days of transfection with 2 different siRNA for *DHODH* or treated with 0.01 μM or 0.1 μM brequinar for 24 h. Fresh media with drugs were replaced the day after siRNA transfection or 24 h before harvesting for the brequinar-treated cells. (E) Amounts of brequinar, UMP, UDP, UTP and uridine in the LN229 measured by LC-MS/MS after 3 days of transfection with 2 different siRNA for *DHODH* or treated with 0.01 μM or 0.1 μM brequinar for 24 h. Fresh media with drugs were replaced the day after siRNA transfection or 24 h before harvesting for the brequinar-treated cells. Numerical values for each of the experiments represented are available in S8 Data.
(PDF)

**S4 Fig. 5-fluorouracil reduces proliferation but does not decrease pre-rRNA synthesis in glioblastoma cells.** (A) Schematic representation of the action of the pyrimidine inhibitors 5-fluorouracil (5-FU, a TS inhibitor), brequinar, and ML390 (DHODH inhibitor). (B) Proliferation curve of increasing amount of 5-FU for the LN229, GBM9 and SF188 cells. Media with drugs were changed every 2 days for 6 days. (C, D) qPCR of 47S pre-rRNA normalized by *ACTIN* mRNA amount in LN229 (C) and SF188 (D) glioblastoma cells with or without 5-FU and with or without uridine for 24 h. 5-FU did not affect the production of pre-rRNA. Media with drugs were changed the day after seeding. (E) Relative proliferation of the SF188 glioblastoma cells with or without 5 μM 5-FU and with or without 25 μM uridine. Proliferation was assessed by crystal violet staining after 4 days of treatment. Uridine did not rescue the effects of 5-FU in proliferation. Numerical values for each of the experiments represented are available in S9 Data.
(PDF)

**S5 Fig. Brequinar limits the production of ribosomal RNA production and tumor growth of glioblastoma xenografts.** (A) Representation of subcutaneous xenograft experiment with LN229 glioblastoma cells. Mice were treated with 15 mg/kg brequinar every 3 days by intraperitoneal injections (IP). (B) Representation of LN229 xenograft tumors from control and brequinar-treated mice with 10 mg/kg by daily IP injections. Tumor weight measurements of LN229 xenografts once the experiment was finished and the tumors harvested are indicated below each tumor picture. (C) Table showing the tumor and mouse weights of each mouse in the subcutaneous xenograft experiment. (D) qPCR of *ACTIN* mRNA levels normalized to same amounts of total RNA in the LN229 subcutaneous xenograft mice tumors. (E) Correlation of pre-rRNA, mature 28S and 18S rRNA, and *ACTIN* RNA levels with tumor weight. Only the pre-rRNA RNA levels in the control group showed significant correlation with tumor size. (F) qPCR of *ACTIN* mRNA levels normalized to same amounts of total RNA in the brain tissue of the mice used for the xenografts experiments in (B). Asterisks indicate p-values ≦0.05. Numerical values for each of the experiments represented are available in S10 Data.
(PDF)

**S6 Fig. The inhibition of *de novo* pyrimidine biosynthesis pathway causes nucleolar stress in glioblastoma cells.** (A) Immunofluorescence of the rDNA transcriptional factor UBF in LN229 cells with or without 0.1 μM brequinar and with or without 100 μM uridine for 24 h. UBF clustered in the edges of the nucleolus (indication of nucleolar stress) upon brequinar treatment, which was rescued by the addition of uridine. Scale bar is 5 μm. (B) Immunofluorescence of UBF in ARPE with or with of 0.1 μM brequinar and with or without 100 μM uridine for 24 h. UBF did not cluster in the edges of the nucleolus upon brequinar treatment. (C) Immunofluorescence of the rRNA processor nucleophosmin 1 (NPM1) in LN229 cells with or without 0.1 μM brequinar and with or without 100 μM uridine for 24 h. NPM1 redistributed

to the edges of the nucleolus (indication of nucleolar stress) upon brequinar treatment, which was rescued by the addition of uridine. (D) Immunofluorescence of NPM1 in GBM9 cells with or without 0.5 μM brequinar and with or without 25 μM uridine for 24 h. NPM1 redistributed to the edges of the nucleolus (indication of nucleolar stress) upon brequinar treatment, which was rescued by the addition of uridine. (E) Immunofluorescence of NPM1 GFP-tagged in LN229 cells with or without 0.1 μM brequinar and with or without 100 μM uridine for 24 h. NPM1 redistributed to the edges of the nucleolus (indication of nucleolar stress) upon brequinar treatment, which was rescued by the addition of uridine. (F) Immunofluorescence of rRNA in LN229 cells by using the anti-rRNA Y10b antibody with or without 0.1 μM brequinar and with or without 100 μM uridine for 24 h. Brequinar decreased the amounts of rRNA, which was rescued by uridine. (G) Immunofluorescence of rRNA in GBM9 by using the anti-rRNA Y10b antibody with or without 0.5 μM brequinar. Brequinar decreased the amounts of rRNA. Right panel shows the quantification by image J. For all the experiments, media with drugs and uridine were replaced the day after seeding. Numerical values for each of the experiments represented are available in S11 Data.
(PDF)

**S1 Table. Brequinar reduces the levels of pyrimidine nucleotides specifically in glioblastoma tumor xenografts tissues.** Table showing the amounts of brequinar, UMP, UDP, UTP and uridine in the LN229 tumor xenografts, brain tissue, liver tissue and serum in each mouse.
(PDF)

**S2 Table. Cell lines, antibodies, siRNA, primers, and reagents used in the present manuscript.**
(PDF)

**S1 Data. Numerical data of experiments shown in Fig 1.** Tables including numerical values of the experiments represented in Fig 1.
(XLSX)

**S2 Data. Numerical data of experiments shown in Fig 2.** Tables including numerical values of the experiments represented in Fig 2.
(XLSX)

**S3 Data. Numerical data of experiments shown in Fig 3.** Tables including numerical values of the experiments represented in Fig 3.
(XLSX)

**S4 Data. Numerical data of experiments shown in Fig 4.** Tables including numerical values of the experiments represented in Fig 4.
(XLSX)

**S5 Data. Numerical data of experiments shown in Fig 5.** Tables including numerical values of the experiments represented in Fig 5.
(XLSX)

**S6 Data. Numerical data of experiments shown in S1 Fig.** Tables including numerical values of the experiments represented in S1 Fig.
(XLSX)

**S7 Data. Numerical data of experiments shown in S2 Fig.** Tables including numerical values of the experiments represented in S2 Fig.
(XLSX)

**S8 Data. Numerical data of experiments shown in S3 Fig.** Tables including values of the experiments represented in S3 Fig.
(XLSX)

**S9 Data. Numerical data of experiments shown in S4 Fig.** Tables including values of the experiments represented in S4 Fig.
(XLSX)

**S10 Data. Numerical data of experiments shown in S5 Fig.** Tables including values of the experiments represented in S4 Fig.
(XLSX)

**S11 Data. Numerical data of experiments shown in S6 Fig.** Tables including values of the experiments represented in S6 Fig.
(XLSX)

## Acknowledgments

We are grateful to the Sorrell lab members and Dr. Sandra Schmid, Dr. Peter Michaely, Dr. Javier León Serrano, and Dr. Marcos Iglesias Lozano for their valuable input and the Live Cell Imaging Facility at UTSW directed by Dr. Kate Luby-Phelps.

## Author Contributions

**Conceptualization:** M. Carmen Lafita-Navarro, Maralice Conacci-Sorrell.

**Data curation:** M. Carmen Lafita-Navarro.

**Formal analysis:** M. Carmen Lafita-Navarro.

**Funding acquisition:** Maralice Conacci-Sorrell.

**Investigation:** M. Carmen Lafita-Navarro, Niranjan Venkateswaran, Jessica A. Kilgore, Suman Kanji, Jungsoo Han, Spencer Barnes, Noelle S. Williams, Michael Buszczak, Sandeep Burma, Maralice Conacci-Sorrell.

**Methodology:** M. Carmen Lafita-Navarro, Niranjan Venkateswaran, Jessica A. Kilgore, Suman Kanji, Jungsoo Han, Spencer Barnes, Noelle S. Williams, Michael Buszczak, Sandeep Burma.

**Project administration:** Maralice Conacci-Sorrell.

**Software:** M. Carmen Lafita-Navarro.

**Supervision:** Maralice Conacci-Sorrell.

**Validation:** M. Carmen Lafita-Navarro.

**Visualization:** M. Carmen Lafita-Navarro.

**Writing – original draft:** M. Carmen Lafita-Navarro, Maralice Conacci-Sorrell.

**Writing – review & editing:** M. Carmen Lafita-Navarro, Michael Buszczak, Sandeep Burma, Maralice Conacci-Sorrell.

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
