## [Decision Letter · Decision Letter 0]

21 Jun 2020

Dear Dr. Sorrell

Thank you very much for submitting your Research Article entitled 'De novo pyrimidine biosynthesis is necessary to maintain ribosomal RNA transcription in glioblastoma' to PLOS Genetics. Your manuscript was fully evaluated at the editorial level and by independent peer reviewers. The reviewers appreciated the attention to an important problem, but raised some substantial concerns about the current manuscript. Based on the reviews, we will not be able to accept this version of the manuscript, but we would be willing to review again a much-revised version. We cannot, of course, promise publication at that time.

If you decide to revise the manuscript for further consideration at PLOS Genetics, please aim to resubmit within the next 60 days, unless it will take extra time to address the concerns of the reviewers, in which case we would appreciate an expected resubmission date by email to plosgenetics@plos.org.

[LINK]

We are sorry that we cannot be more positive about your manuscript at this stage. Please do not hesitate to contact us if you have any concerns or questions.

Yours sincerely,

Simon Khagi, M.D.

Guest Editor

PLOS Genetics

Peter McKinnon, Ph.D.

Section Editor: Cancer Genetics

PLOS Genetics

Reviewer's Responses to Questions

**Comments to the Authors:**

Reviewer #1: Lafita-Navarro et al. report that glioblastoma cells, sensitive and resistant to the chemotherapeutic agent temozolomide (TMZ), display strong sensitivity to the inhibition of de novo pyrimidine synthesis. The authors demonstrate that the inhibition of the mitochondrial pyrimidine enzyme, DHODH, impairs the synthesis of ribosomal RNA and leads to nucleolar stress. Furthermore, pyrimidine depletion relocalizes the distribution of nucleolar proteins specifically in transformed cells but not in normal cells. DHODH inhibition decreases pyrimidine levels and the growth of glioblastoma tumor cells and interestingly decreases DNA repair mechanisms enabling the prevention of resistance mechanisms induced by TMZ.

Fundamentally, this study is not conceptually novel because of the emerging literature in the cancer metabolism field, demonstrating the potential therapeutic strategy of targeting DHODH in several cancers. However, this study sheds light on the role of pyrimidines in glioblastoma and is interesting from a translational standpoint.

Major comments:

1. This study is well-executed and provides interesting translational information on the impact of targeting pyrimidine synthesis in GBM. The authors demonstrate that the immature 47S pre-rRNAs are controlled by pyrimidine availability. The specificity of this regulation to pyrimidine levels seems arbitrary as it is also predictable that purine depletion should have the same impact on newly synthesized rRNA.

2. While the treatment with Brequinar led to an increase in p53 levels, TMZ does not seem to affect p53. The mechanisms by which Brequinar induces a decrease in cancer cell proliferation are probably dependent on the tumor-suppressive functions of p53. The authors should measure the levels of p21, the canonical target of p53, and measure the cytostatic effects of TMZ and Brequinar on the GBM cell cycle.

3. The absence of pyrimidine depletion in the brain tissue of the mice in response to Brequinar treatment is remarkable. However, it raises the question of whether Brequinar spares other nonproliferative tissues. The authors should present the effects of Brequinar on other differentiated tissues. Even though brain cells uptake detectable amounts of Brequinar, it is intriguing that there are no significant effects on the steady-state levels of pyrimidine nucleotide. This result suggests that most of the brain cells depend on the pyrimidine salvage pathway rather than the de novo pathway to maintain their nucleotide pool.

4. The combination of TMZ with Brequinar does not show any synergistic or additive effects. However, the model presented in Figure 5 suggests otherwise. If Brequinar decreases the synthesis of pyrimidines and the mechanisms of DNA repair (MGMT), and that TMZ activates MGMT to trigger DNA repair, then the combination should improve the antiproliferative effects.

5. The decrease in GBM LN229 tumor vascularization induced by Brequinar treatment is intriguing. The authors showed that VEGFA expression is reduced in response to Brequinar. Does pyrimidine depletion also decrease HIF1-a expression in tumors?

Minor comments:

Fig S2G: Thymidylate synthase converts dUMP to dTMP. The schematic should be amended, as “TMP” should be dTMP, dTDP etc. through the rest of the salvage pathway.

Reviewer #2: This study demonstrates that DHODH inhibitor Brequinar has a selective cytostatic effect on glioblastoma cells that rely extensively on de-novo pyrimidine synthesis. The authors hypothesize that the decrease in cell proliferation and/or viability of GBM cells exposed to Brequinar was due, at least in part, to decrease in ribosomal RNA production and nucleolar stress.

Overall, in my opinion, this hypothesis is supported by the data. I have the following suggestions to improve the manuscript.

1. It would be good to ensure that this phenotype was not an off-target effect of Brequinar, by reproducing these key findings with knocking down the endogenous DHODH in these cells. In the current version of the manuscript, the authors examined the effects of DHODH knockdown only on the cell number. They can also examine whether DHODH knockdown causes nucleolar stress similar to Brequinar by UBF and NPM1 labeling. In addition, the amount of rRNA in DHODH knockdown can be measured by qPCR and Y10b immunofluorescence. The effectiveness of DHODH siRNA knockdown (Fig. S2D) needs to be verified by western blotting.

2. Measurements of pre-processed rRNA (47S) and total (18S and 28S) by qPCR: the description of qPCR analysis is quite sparse and does not contain important details such as the number of replicates and the analysis method. For instance, why is the Y scale (relative RNA level) different in Fig. 1 E-F versus Fig. 3 G and J; how were these data normalized? In addition, I could not find primer sequences in the current version of the manuscript.

3. Crystal violet staining quantifications were used in multiple experiments to infer relative proliferation (Fig. 2A, C,E,F,J,K; S2A,B,D,E,F,H,K,L). How was the crystal violet staining quantified (i.e. absorbance, manual colony counting, something else), and how was it normalized? Moreover, it is hard to interpret whether the effects of drug treatments were cytostatic or cytotoxic (i.e. causing a decrease in cell proliferation and/or cell death).

4. Y10b and p53 immunofluorescence labeling (Fig. 4 C-D) also needs to be quantified.

5. Western blots in Fig. 2G-H and Fig. 2L can also use quantification of gamma-H2AX and p53 levels.

Reviewer #3: Recent high-impact studies have drawn attention to the vulnerability of glioblastoma to inhibition of de novo pyrimidine synthesis (Refs. 10, 11). Consistent with that, the authors report that glioblastoma cells have active de novo pyrimidine biosynthesis. They show brequinar inhibition of DHODH in that pathway in GBM results in inhibition of ribosomal RNA synthesis and induces nucleolar stress. This treatment reduces glioblastoma proliferation, including in cells made resistant to the chemo agent temozolomide in vitro. The authors thus propose pyrimidine biosynthesis inhibition may resolve chemo resistance in GBM.

The writing is uniformly excellent and concise. The observation of reduced vascularization in xenograft tumors with brequinar seems interesting.

My overarching concern is that the study is built on previous findings implicating de novo pyrimidine biosynthesis being an Achilles heel in GBM rather than, say, performing genomic experiments (or data mining) to nominate pathways or prioritize that for pyrimidine. There is weak or no justification for all the models used as stand-ins for human GBM, and different cells are used for different experiments with different controls without justification. Whereas all of us in this field struggle with the balance between biological relevance vs. effect size/penetrance (feasibility), this work seems to ignore that without accounting for it. With the possible exception of the mRNA expression data mining, there is little evidence this work is defining general aspects of GBM biology in cell culture, xenografts or natural cancer.

The selectivity of the primary agent is suggested to be supported by a second inhibitor of the same target, but the evidence is not convincing. They created a drug resistance model, but it is not clear to me if what they achieved is valid in the context of human chemo resistance.

The back and forth between proliferation, apoptosis, p53, DNA repair, etc. was confusing. I have no problem thinking about their complex interconnections, but there should first be some context and explanation of the relevance of each and how they are being interpreted to interact. Here these aspects seemed to be used almost interchangeably to make one major argument.

Major concerns

1. It is not clear how the analysis of mRNA expression was done (e.g., what normalization approach, which algorithm).

2. Methods and statistics were very, very minimally described. For example, I couldn’t find descriptions or sources of any of the antibodies.

3. Three cell lines are used as cell culture GBM models. Why these cell lines? On what basis was one of those chosen for the xenograft studies? What is known genetically about these, particularly oncogenic pathways? In addition to refs., a suppl. table of that would be useful.

4. What was rationale for that control cell (normal human p14ARF-/- immortalized) vs., say, stage I, II and III glioma?

5. Re:previous two items: whether it was convenience or absolute necessity that resulted in chosen cell lines and controls, the authors should provide rationale for their study design, up front in Intro or Results.

6. There is no description of western blotting methods, including signal detection. The signal is much lower for the control actin mRNA in the control cell (normal human p14ARF-/- immortalized) and saturated for the test cells.

7. It was claimed that they “confirmed that DHODH and UMPS protein levels were higher in GBM cell lines (LN229, GBM9 and SF188) compared to normal human p14ARF-/- immortalized”. The latter showed no expression of the test mRNAs. This suggests the proper experiment would be different stages of glioma. I don’t object to doing such experiments, but the information has little usefulness without more context.

8. What was rationale for using retinal epithelial ARPE cells as controls?

9. Cell culture concentrations of brequinar and ML-390 were determined empirically here, but this is a critical issue that would be important to compare to other cell line publications, especially the latter which only shows an effect in uM concentration.

10. Suppl. data do not show experiments and results clearly stated in Results. E.g. 1, “First, we measured the proliferation of non-transformed human p14ARF-/- astrocytes, and retinal epithelial ARPE cells as controls, and the GBM cells LN229, GBM9, and SF188 over 6 days in the presence of increasing amounts of brequinar (Fig 2A)”. But S2A did not have GBM9 data.

11. [#8 Cont’d]. E.g. 2: “GBM cells were more sensitive to brequinar compared to ML-390, based on activation of apoptosis measured by cleaved caspase 3 and PARP1 (S2C Fig).” But Fig S2C does not show data for GBM9 and does not compare LN229 and SF188 to any other cell. More problematic, the untreated cell cleaved PARP1 and Casp3 look very similar to 2uM ML-390 treated LN229 and SF188. The sensitivity of GBM to brequinar and ML-390 is said to be higher but that is not shown for GBM9/ML-390 and the effect of GBM9/brequinar is intermediate to that for the other GBM cells and astrocytes. That suggests there is not a general GBM-pyrimidine pathway association, and, if there is, GBM9 is anomalous. And it raises questions about the in vitro efficacy of ML-390 (this is especially surprising to leave an open question when the investigators measured levels of UMP, UDP, UTP and uridine in the cells by LC-MS/MS for brequinar).

12. There are several problems. It is suggested brequinar and ML-390 somehow cross-validate specificity of pharmacological effects, but they don’t do so convincingly. It is suggested proliferation is inhibited but mechanistic evidence is apoptosis; and is not convincing (preceding comment).

13. It was stated that p53 levels did not increase with TMZ but did with brequinar. However, Fig 2G shows p53 increased with TMZ in LN229, but the figure lacks statistical analysis. That is also problematic because I could not find anywhere those 2G/H experiments were performed, say, in triplicate or were replicated in multiple experiments.

14. The 2G/H experiments again show there is not a general GBM response to TMZ or brequinar according to the p53 effects in LN229 vs GBM9. I do not mean to suggest there should be a general GBM response. However, the authors repeatedly suggest that. It is problematic that a total of 3 cell lines are put forward to show this is not a single cell line study, but different experiments only show data for one or different combinations of two cell lines. They are compared to different controls for different experiments.

15. The idea of generating in vitro chemo resistant cells is interesting. However, it is not clear to me what was done and why. Fig. 2J allows comparison of unselected vs. resistantized SF188. It shows untreated resistant SF188 has 50% of proliferation rate vs. unselected. However, in Fig. 2K, all experiments for unselected and resistant show the untreated had a relative proliferation of 1.

16. One of the interesting observations is that Fig. 2L shows brequinar induces p53 levels, if mildly, in both unselected and resistant SF188. However, there are no statistics given, nor any statement I saw the experiment was done in replicates or repeated multiple times.

**Have all data underlying the figures and results presented in the manuscript been provided?**

Reviewer #1: Yes

Reviewer #2: Yes

Reviewer #3: Yes

PLOS authors have the option to publish the peer review history of their article (what does this mean?). If published, this will include your full peer review and any attached files.

Reviewer #1: No

Reviewer #2: No

Reviewer #3: No

---

## [Editor Report · Decision Letter 1]

14 Sep 2020

Dear Dr Conacci Sorrell,

We are pleased to inform you that your manuscript entitled "Inhibition of the de novo pyrimidine biosynthesis pathway limits ribosomal RNA transcription causing nucleolar stress in glioblastoma" has been editorially accepted for publication in PLOS Genetics. Congratulations!

Yours sincerely,

Simon Khagi, M.D.

Guest Editor

PLOS Genetics

Peter McKinnon

Section Editor: Cancer Genetics

PLOS Genetics

Comments from the reviewers (if applicable):

**Data Deposition**

http://datadryad.org/submit?journalID=pgenetics&manu=PGENETICS-D-20-00812R1

**Press Queries**

---

## [Editor Report · Acceptance letter]

26 Oct 2020

PGENETICS-D-20-00812R1 

Inhibition of the *de novo* pyrimidine biosynthesis pathway limits ribosomal RNA transcription causing nucleolar stress in glioblastoma cells 

Dear Dr Conacci Sorrell, 

We are pleased to inform you that your manuscript entitled "Inhibition of the *de novo* pyrimidine biosynthesis pathway limits ribosomal RNA transcription causing nucleolar stress in glioblastoma cells" has been formally accepted for publication in PLOS Genetics! Your manuscript is now with our production department and you will be notified of the publication date in due course.

With kind regards,

Jason Norris

PLOS Genetics

On behalf of:
